# Loss of function of chromatin remodeler OsCLSY4 leads to RdDM-mediated mis-expression of endosperm-specific genes affecting grain qualities

Avik Kumar Pal, Saumyapriya Rana[ID], Riju Dey[ID], Padubidri V. Shivaprasad[ID]*

National Centre for Biological Sciences, Tata Institute of Fundamental Research, GKVK Campus, Bangalore, India

* shivaprasad@ncbs.res.in

## Abstract

In plants, RNA-directed DNA methylation (RdDM) sequence-specifically targets transposable elements (TEs) and repeats, often in a tissue-specific manner. In triploid endosperm tissue, RdDM also acts as a parental dosage regulator, mediating spatio-temporal expression of genes required for its development. It is unclear how RdDM is initiated and established in endosperm. Rice endosperm-specific imprinted chromatin remodeler OsCLSY3 recruits RNA polymerase IV to specific genomic sites for silencing and optimal gene expression. Here we show that, in addition to OsCLSY3, ubiquitously expressed OsCLSY4 is also crucial for proper reproductive growth and endosperm development. Loss of function of OsCLSY4 led to reproductive and nutrient-filling defects in endosperm. Using genetic and molecular analysis, we show that both OsCLSY3 and OsCLSY4 play overlapping and unique silencing roles in rice endosperm, by targeting specific and shared genomic regions such as TEs, repeats and genic regions. These results indicate the importance of optimal expression of two OsCLSYs in regulating endosperm-specific gene expression, genomic imprinting and suppression of specific TEs. Results presented here provide new insights into the functions of rice CLSYs as upstream RdDM regulators in rice endosperm development, and we propose that functions of their homologs might be conserved across monocots.

## Author summary

The endosperm performs a unique physiological function similar to that of mammalian placenta, providing nourishment to the developing plant embryo and acting as a signaling hub. This tissue is distinct from the other plant tissues in its ploidy levels, parental genome dosage and chromatin architecture. The role of

**Data availability statement:** o All raw and processed sequencing data generated in this study have been submitted to the NCBI Gene Expression Omnibus (GEO; https://www.ncbi.nlm.nih.gov/geo/) under accession numbers GSE290279. Other publicly available data used in this study can be found under GSE229961, GSE180457, GSE138705, GSE130168, GSE229959, GSE215853, GSE130166, GSE215854, GSE158710, GSE131319. The reads were aligned to IRGSP1.0 genome. Source data are provided with this paper.

**Funding:** PVS is supported by NCBS-TIFR (https://www.ncbs.res.in/), Government of India. PVS is also supported by the Department of Atomic Energy, Government of India (https://dae.gov.in/), under Project Identification No. RTI 4006 (1303/3/2019/R&D-II/DAE/4749 dated 16.7.2020). The funders had no role in study design, data collection and analysis, decision to publish, or preparation of the manuscript.

**Competing interests:** The authors have declared that no competing interests exist.

RdDM in regulating tissue-specific epigenomes in plants by silencing TEs and repeats is well known, however, its role in monocot endosperm development and its genome regulation are less understood. In this study, we identified a chromatin remodeler *OsCLSY4* as a key factor involved in rice endosperm development. We found that both *OsCLSY4* and *OsCLSY3* act redundantly and uniquely in specific TE sites to silence them, thereby regulating genes required for endosperm development. We propose similar roles for CLSY3/4 clade members among other cereal crops.

## Introduction

In addition to the embryo, angiosperm seeds contain a protective seed coat and a nourishment tissue called endosperm [1,2]. Unlike gymnosperms, a majority of the angiosperm seeds contain a triploid endosperm which is also a fertilization product, similar to that of the embryo [2–4]. Flowering plants undergo double fertilization events, where one of the two sperm cells fertilizes the egg cell to generate diploid embryo and the other sperm cell fertilizes pre-fused 2n central cell to produce the triploid endosperm [5]. The incorporation of the paternal genome into the endosperm may have enhanced nutrient transfer to the growing embryo, potentially contributing to the rapid diversification and success of flowering plants [6,7].

In cereal seeds, the starch-filled endosperm serves as a major food source for humans and other animals [3,8]. Its parental-genomic dosage, ploidy and chromatin organization are unique when compared to other plant tissues. The molecular mechanism of endosperm development is not yet fully understood [8]. Besides acting as a nourishment-tissue, endosperm also senses environmental changes, abiotic and biotic factors to regulate seed germination [9–12]. Multiple studies in the model plant *Arabidopsis* indicated that hormonal, genetic and epigenetic pathways tightly regulate gene expression during endosperm development [2,3].

Among these mechanisms, epigenetic modifications such as DNA methylation, demethylation and histone modifications are important because they also regulate maternal and paternal genome dose in endosperm [3,13–17]. Several key players in these pathways are well-known in *Arabidopsis* [17–21]. These include players involved in DNA methylation, small RNA (sRNA) biogenesis and mediators of epigenetic modifications. Cytosine methylation which is observed in CG, CHG and CHH contexts (where H corresponds to A, T, or C) in plants requires various methyltransferases such as METHYLTRANSFERASE1 (MET1) and CHROMOMETHYLASE3 (CMT3), that catalyze CG and CHG methylation, respectively. The CHH methylation is mainly mediated by a *de novo* DNA methyltransferase named DOMAINS REARRANGED METHYLTRANSFERASE2 (DRM2), which is guided by sRNAs. In many selective loci, CHH methylation is also regulated by CHROMOMETHYLASE2 (CMT2) [22]. An RNA-mediated regulatory loop is required for initiating and maintaining DNA methylation. A plant-specific DNA-dependent RNA polymerase IV (Pol IV) generates short single-stranded (ss) RNA

transcripts predominantly from TEs and repeats. These ssRNAs get converted into double-stranded (ds) forms by RNA DEPENDENT RNA POLYMERASE2 (RDR2) and further processed into 24 nt sRNA duplexes by DICER-LIKE3 (DCL3). The 24 nt sRNAs are preferentially loaded onto ARGONAUTE4/6/9 (AGO4/6/9) to guide DRM2 to methylate TEs and repeats by associating with long non-coding transcripts generated by plant-specific DNA-dependent RNA polymerase V (Pol V). Recruitment of Pol IV at specific loci requires a family of putative SNF2 chromatin remodelers named CLASSYs (CLSY), sometimes along with SAWADEE HOMEODOMAIN HOMOLOG1 (SHH1) proteins. The tissue-specific expression pattern of CLSYs regulate tissue-specific DNA methylation in *Arabidopsis* [23–28]. Mutations in these genes affect global DNA methylation levels, thereby resulting in abnormal growth and development [21,29–33].

Most of the mechanistic understanding about CLSY proteins is derived from *Arabidopsis* that codes for four CLSY proteins. These are major upstream regulators of the RdDM pathway. The *clsy* quadruple mutant shows global loss of 24 nt sRNAs, similar to *pol iv* mutant [26,34]. The sRNA production and DNA methylation were affected in single CLSY mutants, suggesting that they are largely non-overlapping and locus-specific regulators [26,27]. However, *clsy1,2* and *clsy3,4* double mutants affected a greater number of sRNA loci than single mutants, suggesting some degree of redundancy within the CLSY family proteins [26,27,35]. *Arabidopsis* CLSY4 was identified as a DNA demethylation factor and strong hypermethylation at several loci was observed in *clsy4* [36]. Single *clsy* mutants displayed many hypermethylated DNA loci along with RdDM-dependent hypomethylated sites, the mechanistic understanding of which was not understood [32,36]. Unlike *Arabidopsis*, CLSY mis-expression lines in rice displayed partial or complete sterility due to abnormal reproductive development and seed-filling defects [32,37].

Endosperm development is quite distinct in dicots compared to monocots. In mature dicot seeds, endosperm tissue gets used up during embryo development, but in monocots, endosperm is retained and utilized during germination. However, key regulators of endosperm development among monocots are not fully identified yet. While the RdDM pathway seems to play an important role in rice reproduction as seen in a few genetic studies [32,33,37–40], a clear mechanistic understanding has been elusive. Detailed expression patterns and domain architectures of the three rice CLSY proteins are known [32,41,42]. *Poaceae*-specific duplications within *CLSY3/4* clades was well documented previously [42]. The *OsCLSY3* (Os02g0650800/LOC_Os02g43460) is an endosperm-preferred gene that regulates a set of imprinted genes in endosperm *via* imprinted OsCLSY3-dependent sRNAs [32]. *OsCLSY1* (Os07g0692600/LOC_Os07g49210) is essential during anaerobic germination [43]. *OsCLSY4* was identified as an important RdDM pathway gene in a genetic mutant screening where it was named as *fem2* [37]. *OsCLSY4* (Os05g0392400/LOC_Os05g32610) is a ubiquitously expressed gene regulating a majority of the TEs in vegetative stages. In *fem2* seedlings, overexpression of *OsCLSY3* (*FEL1*), but not *OsCLSY1* (*FEL2*), partially complemented DNA methylation levels at some specific genomic loci, indicating a partial redundancy in functions between *OsCLSY3* and *OsCLSY4* [37].

In this study, we identified redundant and unique RdDM-associated functions of two *OsCLSY* genes that express in the endosperm, a tissue largely exhibiting DNA hypomethylation. We show that, in addition to OsCLSY3, OsCLSY4 also redundantly regulates RdDM at specific genomic loci in the endosperm. RNA-seq, sRNA-seq and whole genome DNA methylation analysis indicated that both OsCLSY proteins show shared and unique preferences for TEs and repeats. We also identified numerous loci where we observed hypermethylation in the absence of CLSYs. These results indicate that the endosperm-preferred OsCLSY3, as well as the ubiquitously expressed OsCLSY4 are crucial for endosperm development. Significant numbers of seed development, endosperm development, as well as imprinted genes were misexpressed in their knockdown (kd) lines, providing a molecular basis for the fertility/seed development defects observed in transgenic lines. These findings support the presence of an interplay between redundant and parallel epigenetic pathways mediated by OsCLSY3 and OsCLSY4 proteins in mediating the cellularization timing and development of endosperm in rice.

## Results

### OsCLSY4 loss of function led to reduced fertility and increased seed defects

The maternally expressed OsCLSY3 is an important regulator of the endosperm-specific RdDM pathway controlling endosperm-specific gene expression in rice. Surprisingly, its expression was controlled by the ubiquitously expressed OsCLSY4 targeting two MITE TEs for RdDM in its promoter [32]. The presence of many OsCLSY3-independent sRNA loci in the endosperm was attributed to redundancy with other CLSY members [32]. In addition, the hyper-methylation observed at several genomic loci indicated a competition with another CLSY or other players involved in RdDM. In endosperm tissue, OsCLSY4 is also expressed (S1A and S1B Fig) while expression of OsCLSY1 is negligible [32]. We found that both OsCLSY3 and OsCLSY4 possess N-terminal intrinsically disorder regions and C-terminal SNF2 helicase domains, sharing 31.3% sequence identity and 43.4% similarity between themselves. The N-terminal region of these proteins was comparatively less conserved but C-terminal regions (1030–1440 amino acids) were very well-conserved. We also compared OsCLSY4 protein with the CLSY homolog of maize (RMR1) and found that it exhibited around 64.4% sequence identity and 73.4% similarity. In maize, RMR1 is an important RdDM pathway component and a regulator of paramutation [44]. In previous studies, knockout (ko) of OsCLSY3 and OsCLSY4 led to partial or complete sterility [32,37]. All these pointed out to the possibility that OsCLSY4 might also be playing an important role in rice reproductive development.

To understand the functional redundancy or unique roles of OsCLSY3 and OsCLSY4 in rice endosperm, and to compare with osclsy3-kd data, we used an efficient artificial miRNA (amiR) based kd method to silence OsCLSY4 in *indica* rice line Pusa Basmati-1 (henceforth PB1) [45]. We found up to 70% reduction in *OsCLSY4* transcripts in kd transgenic lines in RT-qPCR analysis (Fig 1A). As observed in os*clsy4/ fem2* mutant, these osclsy4-kd plants had reduced vegetative growth (S1C and S1D Fig). Most importantly, osclsy4-kd lines showed many reproductive defects, such as in panicle length, number of spikelets and grain filling rate per panicle, all of which were drastically reduced (Fig 1B and 1C). The RdDM pathway plays a crucial role in pollen viability and reproductive success in plants and several mutants in this pathway, such as *nrpd1a, clsy3*, several AGO genes exhibit pollen sterility [21,32,40]. We performed pollen viability assay and found that the number of viable pollens was also low in osclsy4-kd, indicating possible reasons for the reduced fertility observed (S1E Fig). Similar to osclsy3-kd, the osclsy4-kd seeds were smaller than wild type PB1 seeds (Fig 1D). However, osclsy4-kd seeds were wider than PB1 and osclsy3-kd seeds (Fig 1D and 1E). While both osclsy3-kd and osclsy4-kd led to smaller seeds, grain chalkiness phenotype was observed only in osclsy4-kd endosperms (Figs 1F and S1F). In monocots, endosperm plays a vital role not just in embryo development, but also in seed germination and hence major defects in endosperm results in germination defects [46]. In seed germination assays, we observed slow germination of osclsy4-kd plants (Fig 1G). The emergence and growth of coleoptiles were delayed in osclsy4-kd when compared to control seeds. Our results suggest that OsCLSY4 is also an important regulator of rice reproduction and endosperm development.

### OsCLSY4 regulated a large number of sRNA loci in rice endosperm

In order to identify the role of OsCLSY4 in regulating accumulation of sRNAs in rice endosperm, we performed sRNA sequencing in 20 days-old osclsy4-kd endosperm tissues. More than 90% of an average of 25 million reads mapped to the rice genome (S1 Table). A drastic reduction of 24 nt sRNAs in osclsy4-kd endosperm tissues similar to that of osclsy3-kd lines was observed (Fig 2A). To compare the importance of both OsCLSY3 and OsCLSY4 in endosperm, we counted 20–25 nt sRNAs in both kd genotypes (S1 and S2 Data). The analysis indicated that 23–24 nt size class sRNAs were more drastically reduced in osclsy4-kd when compared to osclsy3-kd, indicating that the majority of 24-nt sRNAs accumulating in endosperm are under OsCLSY4 control (S2A Fig). Also, in osclsy4-kd, we observed 1500 OsCLSY4-dependent gained sRNA loci (S3A Fig). Among all the sRNA size classes of sRNAs that had 5' A showed varying levels of reduction in osclsy4-kd (S2B Fig). To find OsCLSY4-dependent 23–24 nt sRNAs, we performed ShortStack based

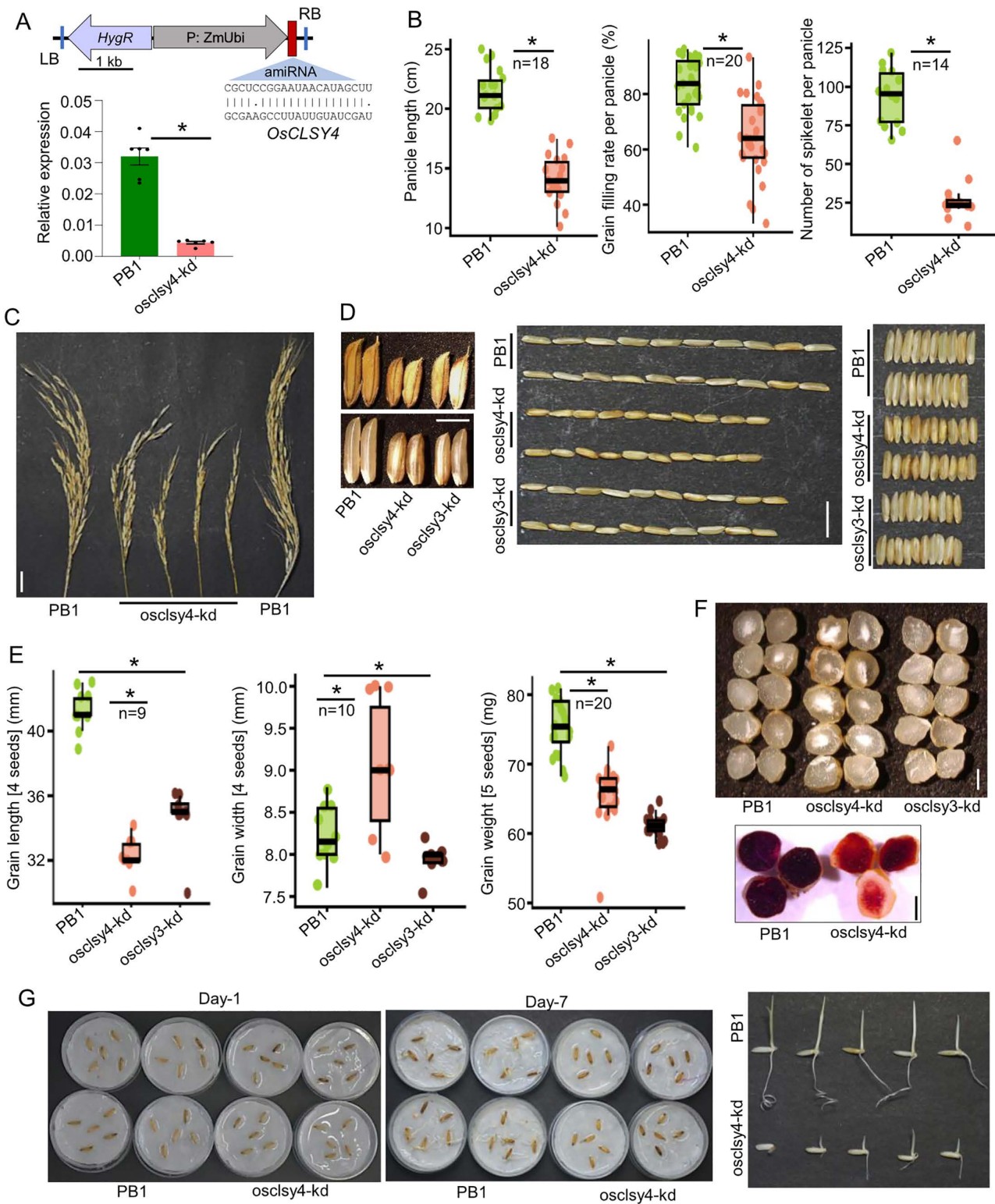

**Fig 1. Endosperm-related phenotypes of osclsy4-kd plants.** (A) T-DNA vector map of amiR construct. The amiR is driven by ZmUbi promoter. RB-right border, LB-left border. Barplot showing levels of OsCLSY4 expression in clsy4-kd endosperm (EN) is shown. OsActin served as internal control. Error bar-SE. *-significant. Two-tailed Student's t-test. (B) Boxplots showing panicle phenotypes. In box plots, *-significant (two-tailed Student's t-test).

(C) Images showing panicle morphology. SB (Scale Bar)-2 cm. (D) Images showing morphology of osclsy3-kd and osclsy4-kd seeds. SB-1 mm. Length (middle panel) and width (right) of 10 seeds across lines are shown. (E) Boxplots showing seed length, width and weight in osclsy4-kd and osclsy3-kd. (F) Seed-phenotype of osclsy4-kd and osclsy3-kd dry grains (top) and iodine-staining (bottom) of osclsy4-kd endosperm sections. SB-0.5 mm. (G) Germination-phenotype of osclsy4-kd. Seeds were germinated and photographed. Individual germinating seeds after 7-days in water (Left) and 5-days in media (Right) are shown, respectively.

analysis and found that the expression of more than 70% of 23–24 nt sRNA loci (10521 CLSY4-dependent loci) that are expressed more than 5 rpm in PB1 were reduced in osclsy4-kd endosperm tissues (Fig 2B). These sRNAs originated from class I (TEs that replicate in genome by a copy-and-paste mechanism) and class II (TEs that replicate *via* cut-and-paste mechanism) TEs, as well as from genic regions (Fig 2C). In agreement with their OsCLSY4 dependency, these sRNA loci were reduced in osclsy4-kd (Fig 2D). On the other hand, 21–22 nt sRNAs from miRNA loci were unaltered in osclsy4-kd endosperm, indicating that CLSY4 majorly regulated sRNAs associated with RdDM (Fig 2E). Major families of TEs such as LINE, LTRs, MITEs and other repeat-derived 23–24 nt sRNAs were significantly reduced in osclsy4-kd lines (Figs 2F and S2C). We found that many OsCLSY4-dependent 23–24 nt sRNAs from TEs overlapped with OsCLSY3-dependent 23–24 nt loci, clearly demonstrating the redundant functions of both OsCLSYs in many TEs (Fig 2G). However, we also observed a reduction in a large number of sRNA loci, particularly in individual CLSY kd lines, which suggested their non-redundant function (Figs 2H and S2D). Together, these results collectively indicate that OsCLSY4 also functions as an RdDM regulator in rice endosperm.

## OsCLSY4 and OsCLSY3 non-redundantly regulate accumulation of sRNAs in endosperm

In *Arabidopsis*, CLSY proteins function both non-redundantly and redundantly in a tissue- and locus-specific manner [26,28]. In order to explore if OsCLSY3 and OsCLSY4 act on specific loci in the endosperm, we analyzed published sRNA datasets derived from osclsy3-kd endosperm tissues (S2 Data) [32]. In osclsy3-kd, we found more than 60% of 23–24 nt sRNA loci reduced when compared to WT control (S2 Table). Along with these OsCLSY3-dependent 23–24 nt sRNA loci, we also obtained 583 loci that were gained in osclsy3-kd (S3B and S3C Fig). Upon overlapping OsCLSY3 and OsCLSY4-dependent 23–24 nt sRNA loci, we obtained 1014 sRNA loci that overlapped with each other ('shared loci), 9507 OsCLSY4-dependent sRNA loci (henceforth, 'OsCLSY4-unique loci') and 3889 CLSY3-dependent loci (henceforth, 'OsCLSY3-unique loci') (Fig 3A and S3 Data).

As expected, OsCLSY4-unique loci were much more reduced in abundance in osclsy4-kd, while OsCLSY3-unique loci were more reduced in osclsy3-kd (Fig 3B and 3C and S3 Data). We observed a marginal reduction of OsCLSY4-unique and OsCLSY3-unique sRNAs in osclsy3-kd and osclsy4-kd endosperm tissues, respectively (Fig 3B and 3C). In order to find out how prevalent shared loci are across TE and other repeat features, we extended the sRNA loci width to 1 kb on both sides (total 2kb extended) and identified a total of 1964 loci that were possibly redundantly regulated by both CLSYs in endosperm tissues (S3D Fig). These results collectively suggested that these rice CLSYs function largely in a non-redundant fashion in endosperm tissues.

Similar to *Arabidopsis*, along with predominant 24 nt sRNAs, there are also 21–22 nt sRNAs in rice endosperm [20,32]. A majority of the 21–22 nt sRNA loci also overlapped with 23–24 nt loci, indicating that their precursors are targeted by multiple DCLs [20]. ShortStack analysis identified 11818 CLSY4-dependent 21–22 nt sRNA loci in rice endosperm (S3E–F Fig). In this analysis, we also identified around 2300 sRNA gained loci in osclsy4-kd (S3E Fig). Around 3987 sRNA loci showed downregulation of both 21–22 and 23–24 nt sRNAs in osclsy4-kd endosperm indicating that OsCLSY4 is an upstream player in these loci (S3G Fig). Similarly, OsCLSY3-dependent 21–22 nt sRNA loci were also identified (S3H–I Fig). A few OsCLSY3-dependent sRNA loci also had sRNAs of both size classes (S3J Fig).

Similar to 23–24 nt sRNAs, there were 21–22 nt sRNAs exclusively regulated by both OsCLSY4 and OsCLSY3. We identified 11301 OsCLSY4-unique and 2591 OsCLSY3-unique 21–22 nt sRNA loci (Fig 3D). As expected,

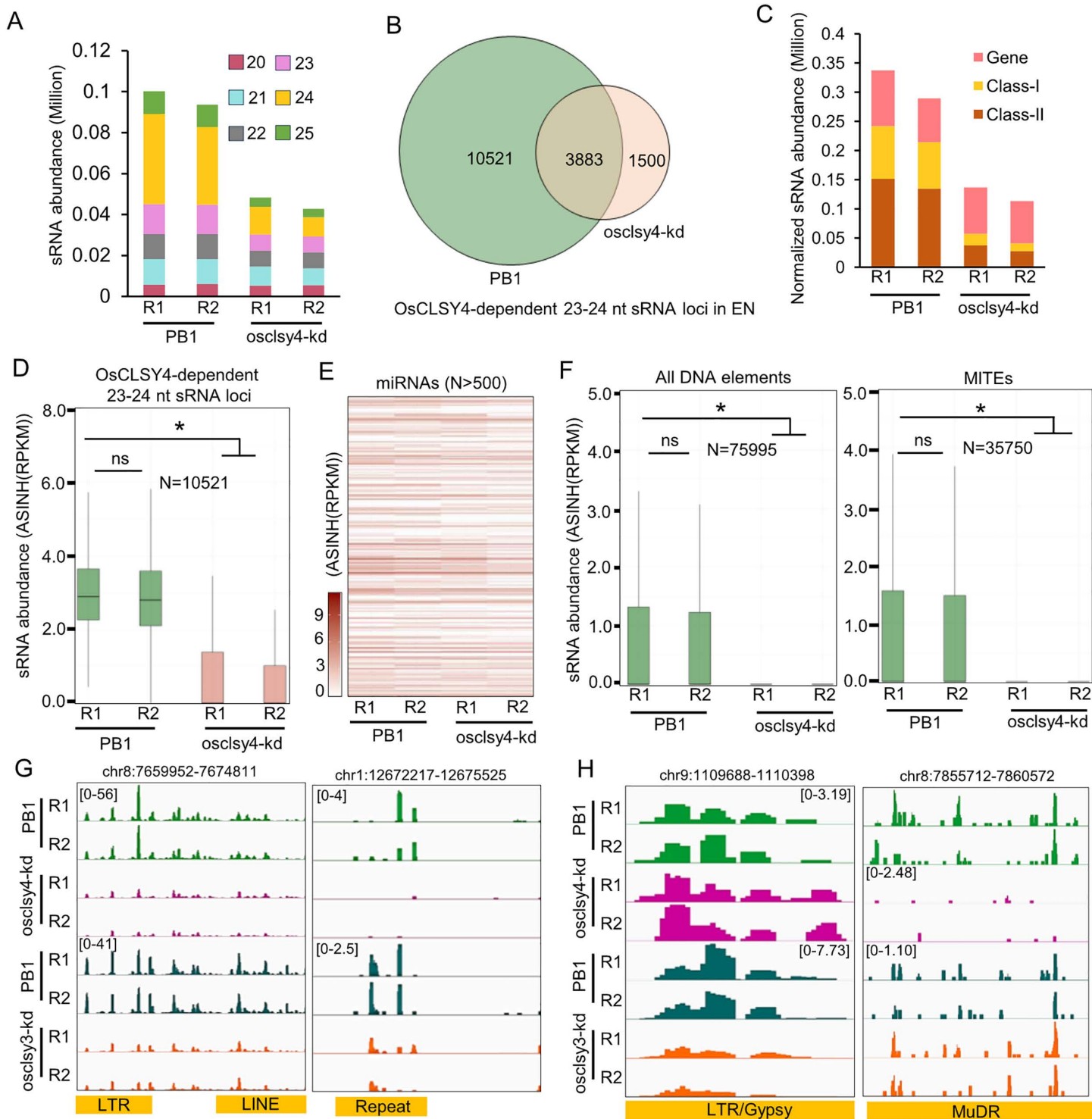

**Fig 2. Endosperm sRNAs are globally reduced in osclsy-kd lines.** (A) Stacked barplot showing abundance of sRNAs (20-25 nt) in PB1 and osclsy4-kd EN tissues. (B) Venn diagram showing 23-24 nt sRNA loci across PB1 and osclsy4-kd EN. (C) Plot showing 23-24 nt sRNA abundance across different genomic features. (D) Boxplot showing abundance of OsCLSY4-dependent 23-24 nt sRNA loci in PB1 and osclsy4-kd EN. *-significant, ns-non-significant (Wilcoxon test p < 0.01). (E) Heatmap showing expression of miRNAs in PB1 and clsy4-kd EN. ASINH converted RPKM values were used for the heat map. (F) Boxplots showing abundance of 23-24 nt sRNAs in different TE types. N-number of loci. (G) and (H) IGV screenshots showing expression of 23-24 nt OsCLSY3- and OsCLSY4-dependent shared (left) and unique sRNA loci (right) respectively.

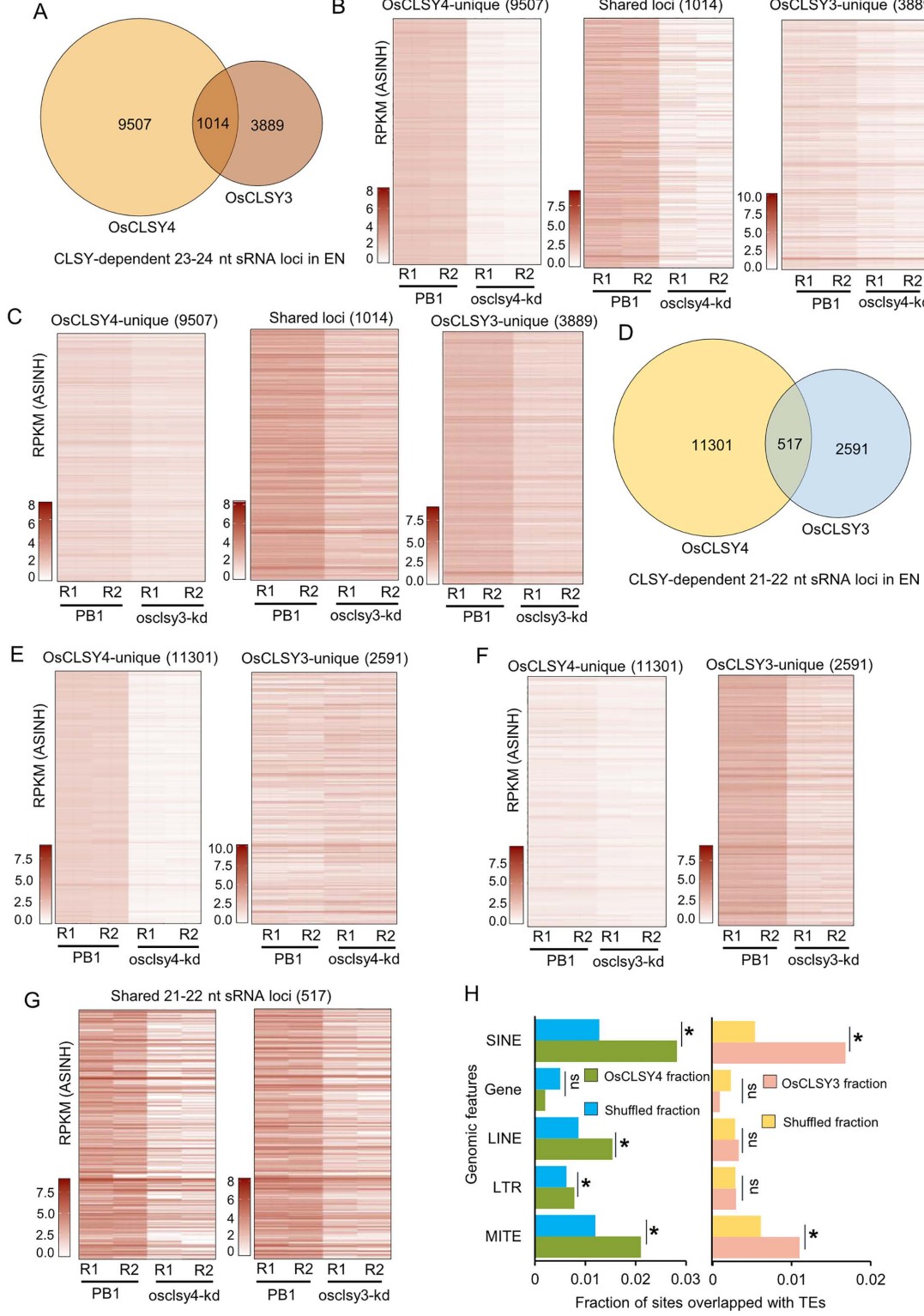

**Fig 3. CLSYs regulate endosperm expressing sRNAs in non-redundant ways.** (A) Venn diagram representing overlap between OsCLSY3 and OsCLSY4-dependent 23-24 nt sRNA loci. (B), (C) Heatmaps showing 23-24 nt sRNA abundance across OsCLSY4-unique, OsCLSY3-unique and shared sRNA loci. ASINH converted RPKM values were used for heatmaps. (D) Venn diagram representing overlap between OsCLSY3 and

OsCLSY4-dependent 21-22 nt sRNA loci. (E), (F), (G) Heatmaps showing 23-24 nt sRNA abundance across OsCLSY4-unique, OsCLSY3-unique and shared sRNA loci, respectively. (H) Bar plots showing fraction of CLSY4-dependent and CLSY3-dependent sRNAs overlapping specific TE features, compared to the fraction obtained from an equal number of shuffled loci across the same TEs. Overlapping regions are measured in base pairs (bp). In plots, *-significant (Fisher's exact test, p < 0.01).

OsCLSY4-unique sRNA loci were downregulated in osclsy4-kd but not in osclsy3-kd (Fig 3E). OsCLSY3-unique sRNA loci were mostly downregulated in osclsy3-kd (Fig 3F). The shared 517 loci were reduced in both kd lines as expected (Fig 3G). Upon extending the length of 21–22 nt sRNA loci, we obtained around 2047 loci in which OsCLSY3 and OsCLSY4 might be functioning redundantly in the endosperm (S3K Fig).

To determine whether CLSYs show preferences for specific genomic features, we compared the CLSY3-and CLSY4-dependent sRNA loci with various genomic features relative to randomly shuffled loci. We observed that OsCLSY4-dependent sRNA loci showed significantly greater overlap with SINE, LINE, MITE and LTRs TEs, whereas OsCLSY3-dependent sRNA loci were more enriched for SINE and MITE elements (Figs 3H and S4A). Interestingly, CLSY4 and CLSY3 show different preference for TEs (S4B Fig). These results indicate that OsCLSY3 and OsCLSY4 regulate distinct sets of sRNA loci in rice endosperm and differ in their preference for specific TE features.

## OsCLSY4 regulates site-specific DNA methylation in the endosperm

CLSYs regulate DNA methylation through the RdDM pathway in both vegetative and reproductive tissues across dicots and monocots [26,36]. Since OsCLSY4 is also expressed in leaves, we performed targeted bisulfite PCR to assess DNA methylation level in osclsy4-kd leaves. We observed specific hypomethylation of the 5S-rDNA repeat in mutant leaves, indicating that OsCLSY4, but not the endosperm-specific OsCLSY3, regulates RdDM in vegetative tissues (S5 Fig).

Further, we performed whole genome bisulfite sequencing of genomic DNA derived from 20-day-old endosperm tissues (S6A Fig). We identified a total of 27345 differentially methylated regions (DMR) in osclsy4-kd in comparison to 27022 DMRs in osclsy3-kd endosperm. In osclsy4-kd, a total of 5984 CG, 10365 CHG and 10997 CHH hypomethylated loci were identified (S4 Data). These numbers were comparable to OsCLSY3-dependent DMRs that we identified previously, indicating that OsCLSY4 also contributes to RdDM in endosperm (S5 Data). Interestingly, we found that 9904 hypo-DMRs were common between osclsy4-kd and osclsy3-kd lines (Fig 4A). Among these shared DMRs, a total of 3626 were exclusively CHH DMRs (Fig 4B). As expected, DNA methylation was reduced in OsCLSY3- and OsCLSY4-DMRs in respective kd lines. However, OsCLSY3 and OsCLSY4-DMRs also exhibited significant reduction in osclsy4-kd and osclsy3-kd lines respectively, indicating that there is redundancy in the activities of these CLSYs in inducing RdDM at some loci (Fig 4C–D). OsCLSY4-dependent sRNA loci overlapped with OsCLSY4-dependent DMRs indicating that OsCLSY4 regulates RdDM at these sites (Fig 4E). Among shared loci, DNA methylation at CHH sites was reduced in both kd lines as expected (Fig 4F). DNA hypomethylation at several TEs, especially MITEs, were observed in osclsy4-kd lines correlating with sRNA reduction (Fig 4E and 4F). These results indicate that OsCLSY4, alongside OsCLSY3, contributes to site-specific DNA methylation in rice endosperm.

## osclsy4-kd plants exhibited pronounced hypermethylation across repeats and genic regions

Previous studies in *Arabidopsis* as well as in rice indicated pronounced hypermethylation at several genomic loci when different CLSY genes were mutated [26,28,32,36]. However, the regions showing hypermethylation were neither shared loci between individual CLSYs, nor they were under the direct control of another CLSY member. While the mechanism of this hypermethylation is unknown, it is possible that more than one factor contributed to this phenomenon. Similar to osclsy3-kd, we observed pronounced hypermethylation at several genomic loci specifically at CHH sites in osclsy4-kd (S6A–B Fig). In osclsy3-kd, 1976 CG, 7930 CHG and 34070 CHH hypermethylated loci were identified using DMRcaller, whereas about 1062 CG, 2617 CHG and 22761 CHH hypermethylated loci were identified in osclsy4-kd lines (S4 and

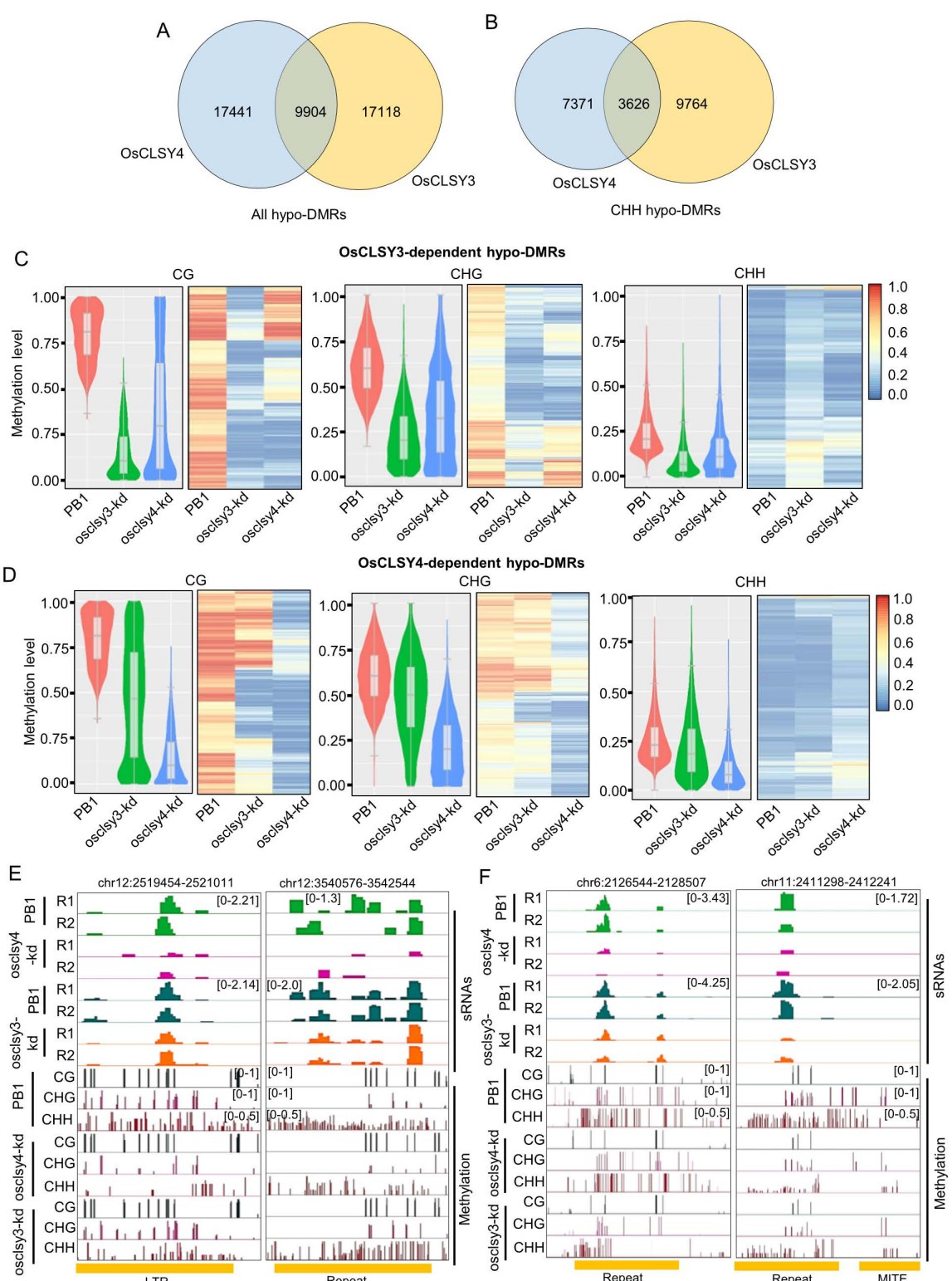

**Fig 4. Both OsCLSY3 and OsCLSY4 regulate methylation through RdDM pathway in endosperm.** (A), (B) Venn-diagrams showing overlap between all hypo-DMRs and CHH context hypo-DMRs in both kd lines in EN, respectively. (C), (D) Violin-plots and heatmaps showing OsCLSY3 and OsCLSY4-dependent hypo-DMRs, respectively. (E), (F) IGV screenshots representing 23-24 nt sRNA and DNA methylation status of CLSY4-unique and shared loci, respectively.

S5 Data and S6C Fig). We overlapped hyper-DMRs between the OsCLSY3 and OsCLSY4 and identified 10021 shared hyper-DMRs across all sites and 7855 shared CHH hyper DMRs (S6A Fig). The hypo- and hyper-DMR regions in both kd lines did not show any preference for any genomic features (S6B Fig).

In our analysis, different categories of TEs, such as MITEs, Gypsy and SINE elements showed DNA hypomethylation mainly in CHG context as expected. However, there were many TEs having hypermethylation, mainly in CHH context (Fig 5A). Among hypermethylated sites, abundance of 21–22 nt sRNAs was not observed in any of the lines, indicating that CHH methylation did not arise from non-canonical 21–22 nt sRNA-mediated DNA methylation (Fig 5B–D). Surprisingly, a majority of the hypermethylated loci did not overlap with any class of sRNAs (Fig 5B–D). In order to understand if osclsy4-kd led to altered expression of components of DNA methylation/demethylation machinery, we tested expression levels of DNA demethylases and methyltransferases and identified that two ROS genes, namely OsROS1a (Os01g0218032) and OsDML4 (Os03g0110800) were downregulated in osclsy4-kd (S7 Data). These results collectively indicate that CLSY proteins might be suppressing DNA methylation and/or DNA demethylation at several sites that lead to hypermethylation at multiple sites in their mutants. It will be interesting to identify whether these alterations are cell-type/tissue-specific, and/or developmentally regulated.

## OsCLSY3 and OsCLSY4 redundantly regulate siren loci in rice

In endosperm tissues, a unique set of sRNA loci called siren (siRNA endosperm-specific) contribute to most of the sRNAs derived from a fewer loci that regulate specific genes involved in development [47–52]. They are predominantly generated from the maternal tissues [49,51,52]. Siren sRNAs act as hybridization barriers by regulating parental dose of gene expression [52]. In *Arabidopsis* and *Brassica rapa*, expression of siren sRNAs are regulated by reproductive-tissue specific CLSY3 and CLSY4 [27,50,51]. We explored if OsCLSY4 is involved in the regulation of siren expression in rice endosperm similar to OsCLSY3 [32]. Among the 797 siren loci in rice, most were reduced in abundance in osclsy4-kd endosperm (S6 Data). These loci were also reduced in osclsy3-kd lines indicating that both these CLSYs regulate expression of siren loci redundantly (Fig 6A–C). As observed in *Arabidopsis* [52], several siren loci were heavily methylated and the status of DNA methylation remained unchanged in osclsy3-kd and osclsy4-kd lines (Fig 6D), however, at selected siren loci, we also observed DNA hypomethylation in both kd genotypes (Fig 6B). A total of 38 siren adjacent genes, such as Os02g0527200, Os10g0177200 and Os01g0629900 were significantly upregulated in kd lines. Also, around 70 siren adjacent genes were downregulated in osclsy4-kd (Fig 6E). Although, the importance of DNA methylation and its contribution to siren RNA loci expression are unknown, it is clear that both OsCLSYs play a major role in their regulation and indirectly the expression of siren loci associated genes. It is important to note that the homologs of rice CLSY3 and CLSY4, i.e., AtCLSY3 and AtCLSY4, also regulate siren loci expression in ovules and endosperm, indicating possible conserved functions of this clade of CLSY members.

## OsCLSY4-dependent sRNA loci control expression of neighboring genes through RdDM

Similar to osclsy3-kd, osclsy4-kd lines also displayed endosperm defects. To understand the molecular basis for the chalkiness phenotype observed in osclsy4-kd, we performed transcriptome analysis of 20-day-old osclsy4-kd endosperms. We documented absence of green-tissue specific marker gene expression indicating that the tissues taken for analysis were free from maternal-tissue contamination (S7A Fig). A total of 1456 upregulated and 1875 downregulated genes were observed in osclsy4-kd (Fig 7A). Although specific pathway genes were not altered, several metabolism-related genes were mis-expressed in osclsy4-kd lines (S7B Fig). Since osclsy4-kd, but not osclsy3-kd, endosperm showed chalkiness phenotypes, indicating a lack of progression to cellularization, we overlapped differentially expressed genes (DEGs) between both kd lines. We identified 2206 and 1628 unique DEGs in osclsy4-kd and osclsy3-kd endosperms, respectively, indicating that both CLSYs play unique roles in endosperm development (Fig 7B). We also identified 30 upregulated and 31 downregulated genes shared between both kd lines (S7C Fig), among which are Os01g0726250 and Os08g0127900

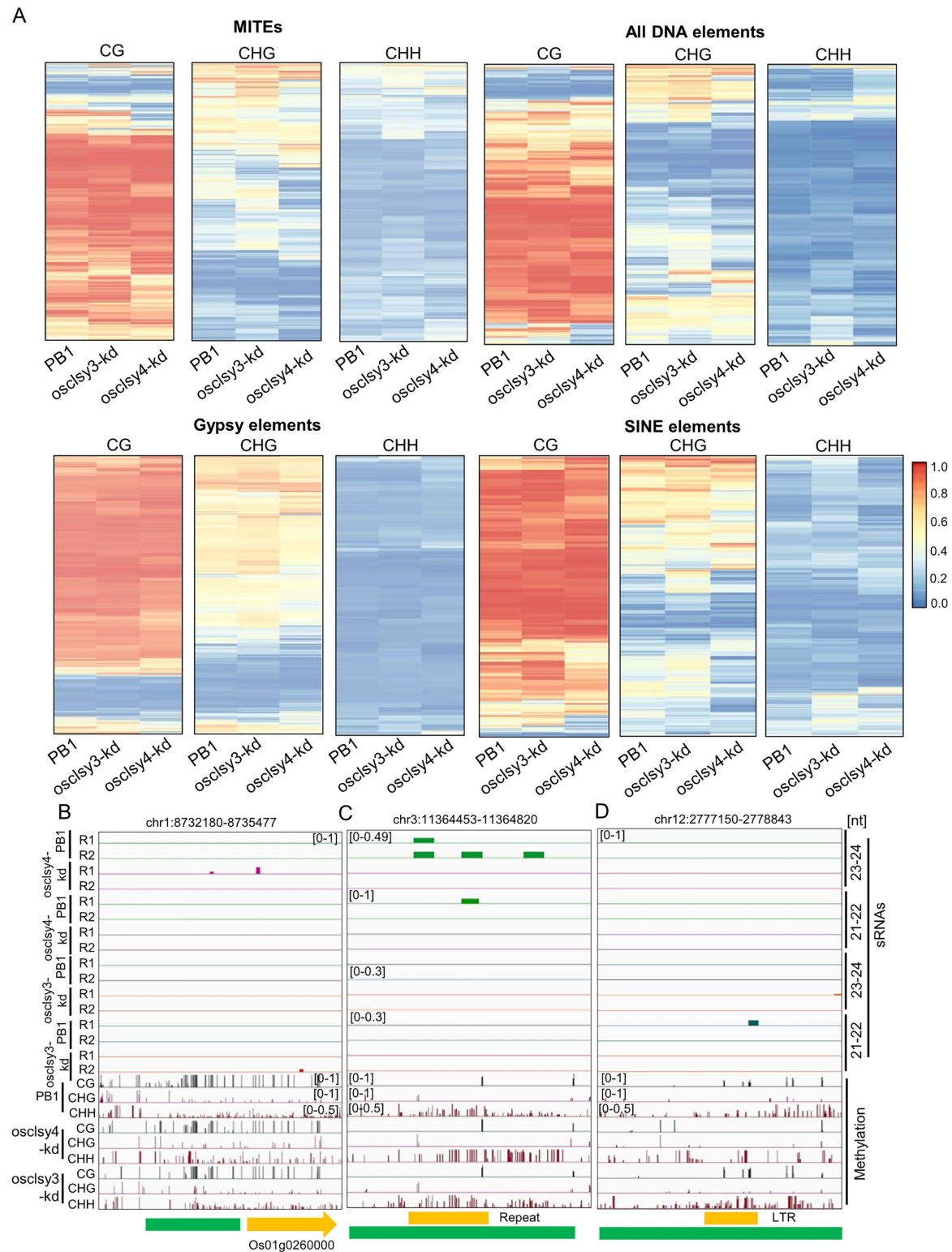

**Fig 5. sRNA independent hypermethylation in osclsy3-kd and osclsy4-kd EN tissue.** (A) Heatmaps showing DNA methylation levels across different TEs in kd lines of both CLSYs in EN. (B), (C), (D) IGV screenshots showing hyper methylated OsCLSY4, shared and OsCLSY3 loci in EN, respectively.

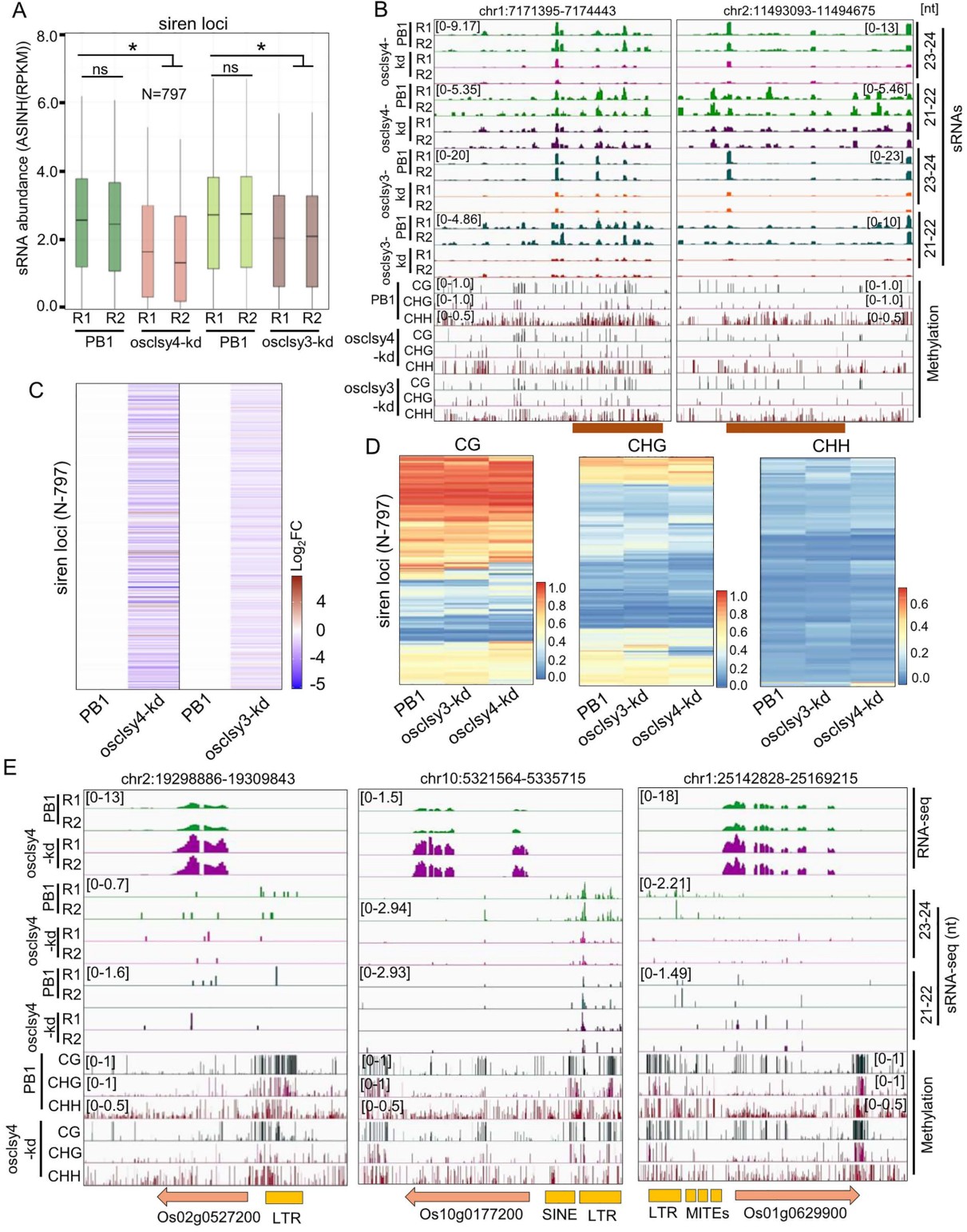

**Fig 6. Both OsCLSYs redundantly regulate siren loci in rice.** (A) Boxplot showing expression of 23-24 nt sRNAs in PB1, and osclsy3-kd and osclsy4-kd EN. *-significant. ns-non-significant (Wilcoxon test p < 0.01). (B) IGV screenshots depicting expression of 23-24 and 21-22 nt sRNAs from siren loci. (C) Heatmaps showing expression of siren loci (N-797) in the osclsy4-kd and osclsy3-kd EN tissues. (D) Heatmaps showing DNA methylation level in siren loci in kd lines EN. (E) IGV screenshots showing expression of siren loci adjacent genes in osclsy4-kd EN.

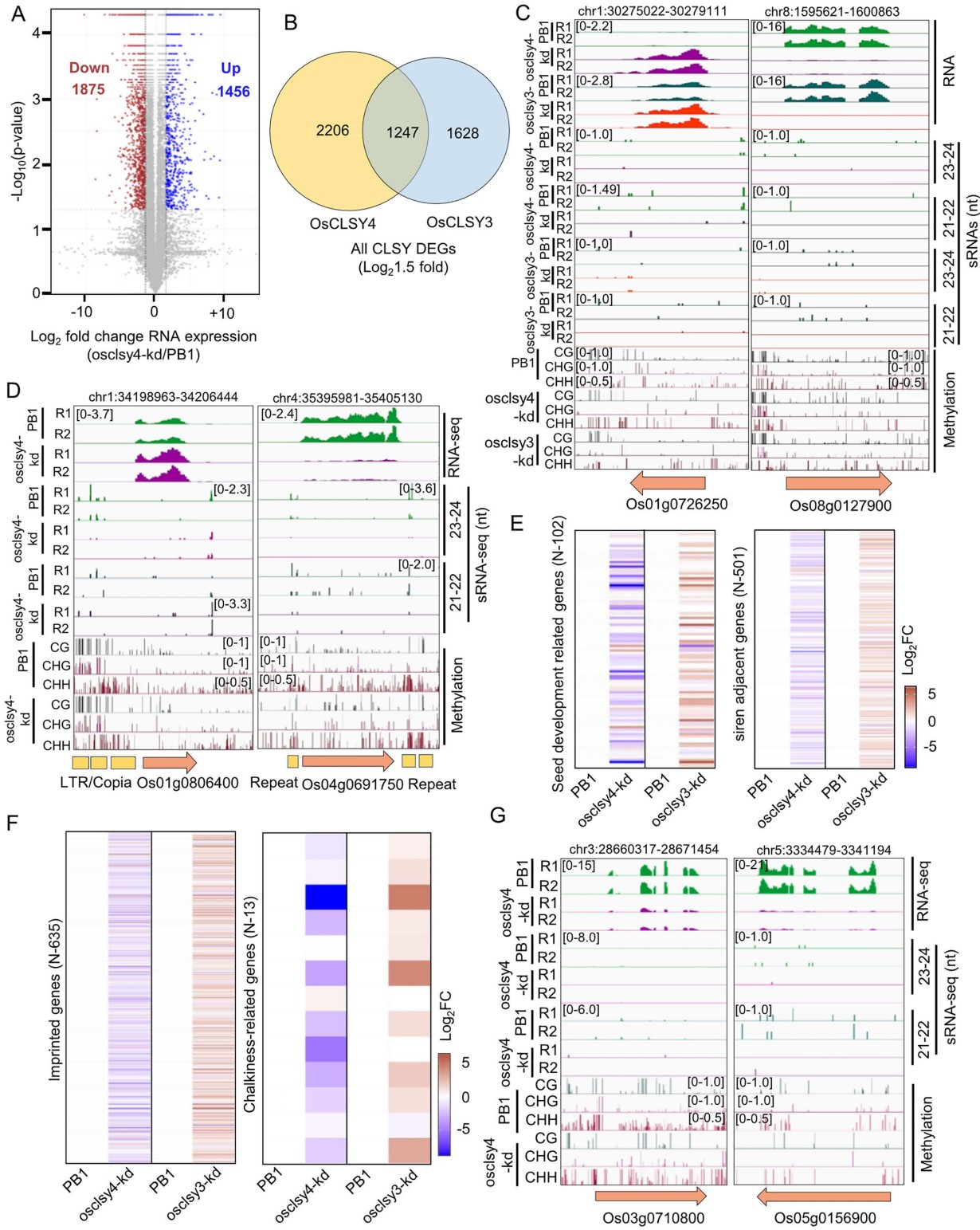

**Fig 7. OsCLSY4 regulates expression of many crucial endosperm-specific genes in rice.** (A) Volcano plot showing expression of genes in osclsy4-kd EN. (B) Venn diagram representing overlap between DEGs (Log2 fold change-1.5) between osclsy3-kd and osclsy4-kd EN. (C) IGV screen shots showing expression of upregulated and downregulated genes in osclsy4-kd and osclsy3-kd EN. (D) IGV screenshots showing upregulated and

downregulated genes in osclsy4-kd EN. (E) and (F) Heatmaps showing expression of seed-development related, siren adjacent, imprinted and chalkiness-related genes in osclsy3-kd and osclsy4-kd EN tissues, respectively. Log2 fold change (FC) RPKM values were used for heatmaps. (G) IGV screenshots showing OsCLSY4-mediated regulation of two chalkiness-related genes in osclsy4-kd EN.

(an endosperm-specific gene 115/ Similar to Globulin 1) (Fig 7C). Genes that showed clear differences between both kd lines included Os07g0568700 (FORAL ORGAN REGULATOR 1) and Os12g0268000 (Cytochrome P450 monooxygenase) that had opposite expression patterns between osclsy3-kd and osclsy4-kd lines (S7D Fig). It is possible that some of the gene expression abnormalities might be due to vegetative and reproductive defects observed in osclsy4-kd.

To explore the role of OsCLSY4-associated RdDM pathway in controlling expression of genes, we analysed the sRNA and methylation levels in the vicinity of upregulated DEGs. The genes such as Os01g0806400 (DUF617) and Os01g0136100 (OsHsp16.9A) showed clear upregulation in osclsy4-kd (Figs 7D and S7E), and their promoters contain TEs and repeats under the control of OsCLSY4. Several seed development-associated genes such as Os11g054650 (glycosyltransferase 6) and Os04g06917500 (phosphoribosylanthranilate transferase) were downregulated in osclsy4-kd mainly due to DNA hypermethylation at their promoters or terminators (Figs 7D and S7E). Since many genes were uniquely mis-expressed between osclsy3-kd and osclsy4-kd endosperms, it is likely that these proteins regulate endosperm development in different stages//cell types.

## OsCLSY4 has distinct targets in vegetative tissues when compared to endosperm

Unlike OsCLSY3, OsCLSY4 is expressed in both vegetative and reproductive tissues in rice. In leaf tissues, OsCLSY4 is the major CLSY member and it appears to target genomic regions such as TEs and repeats [37]. To understand if the sRNA loci regulated in vegetative tissue and the endosperm originate from same regions, we analyzed and compared OsCLSY4-dependent sRNA loci across these tissues (S8 Data). Similar to endosperm tissue, 23–24 nt sRNAs were also reduced in class I, class-II TEs and genic regions in osclsy4-kd (S8A Fig). Interestingly, there were 3544 CLSY4-dependent 23–24 nt sRNA loci overlapping between these tissues. However, a majority of the OsCLSY4-dependent sRNA loci were unique between leaf (69721) and endosperm (6977) tissues, indicating a tissue-specific preference of CLSY4 for genomic loci (S8B Fig). OsCLSY4-dependent sRNA loci originated from TEs and genes in a tissue-specific manner (S8C–S8F Fig). This altered preference of OsCLSY4 might be due to variation in its expression level, possibility of its distinct protein interacting partners or other epigenetic marks. Since CLSY4-dependent sRNAs were mostly different between leaves and endosperm, it is possible that a different set of genes are under its control in these tissues. In agreement with this, several transcription factors specifying the recruitment of AtCLSY3 to initiate RdDM were recently identified, and they had specific roles in male and female reproductive tissues [53,54]. To explore genes that are regulated by OsCLSY4 in two tissues, we compared the DEGs and found only 119 genes that were shared between two tissues (S8G Fig and S9 Data). Together, these results indicate that ubiquitously expressed OsCLSY4 plays a tissue-specific role in regulating sRNA expression and RdDM.

## OsCLSY4 regulates expression of many seed development and chalkiness-related genes in rice

The triploid tissue of endosperm expresses a unique set of genes that are important for the regulation of development, including nutrient accumulation. Endosperm development and growth is a complex process regulated by several signaling, hormonal and epigenetic pathways [55,56]. We observed the mis-expression of many seed development-related genes connected to these pathways in endosperms of both the kd lines (Figs 7E and S8E–S8F). Around 15 seed development-related genes were among the DEGs in osclsy4-kd. Genes such as OsBZR1, OsAGO2, GW6, OsYUCCA11 and NF-YC12 [56–59], that are well-known regulators of endosperm development in rice, were significantly downregulated in osclsy4-kd. Key metabolism-associated genes such as OsGAD2, AWN1 and RAG2 were significantly upregulated in the

osclsy4-kd lines. AWN1 (Os04g0350700) is a negative regulator of seed size and its expression correlated with shorter seeds in osclsy4-kd [60]. Among the MADS-box genes that regulate floral development, cellularization and developmental programmed cell death in rice endosperm [61–63], we observed total of 7 MADS-box genes - OsMADS3, OsMADS4, OsMADS6, OsMADS8, OsMADS17, OsMADS18 and OsMADS29 that were significantly downregulated in osclsy4-kd (S7F Fig). The siren loci adjacent genes were also mis-expressed in osclsy4-kd lines (Fig 7E). Among the 635 imprinted genes in rice, 139 were significantly mis-expressed in osclsy4-kd. Among them, 42 genes had a clear upregulation in osclsy4-kd (Fig 7E and 7F). These results suggest that OsCLSY4 is also an essential and important epigenetic-regulator of gene expression in rice endosperm.

Interestingly, we observed grain-chalkiness phenotype in osclsy4-kd endosperm but not in osclsy3-kd (Fig 1F). The grain-chalkiness is a complex phenotype which arises due to environmental stresses. Defects in starch, protein metabolism, transcription, organelle-development related genes lead to grain-chalkiness [64]. Here, in osclsy4-kd, around 7 well-studied chalkiness-related genes such as OsNUDX7, OsMKK3, OsbHLH96, OsCIN2, OsGF14f, OsCrRLKL3 were significantly downregulated (Fig 7G). The majority of those genes were unaltered upon osclsy3-kd. The two genes OsbHLH96, CHALK5 were also slightly upregulated in osclsy3-kd endosperm (Fig 7F). All these results collectively suggested that CLSY4-mediated RdDM pathway regulates grain-chalkiness in rice.

## Discussion

The endosperm fate differs between monocots and dicots. In monocots such as rice, endosperm is retained and used during germination as a nutrient source [65,66]. Epigenetic pathways such as DNA methylation, demethylation and histone modifications regulate endosperm development. Epigenetic players such as *OsFIE1, OsEMF2a, ZmFIE1* were identified as central to imprinting and endosperm development in monocots [18,67–73]. The RdDM pathway particularly plays a critical role in balancing parental genome contributions and ensuring cellularization timing in endosperm tissues [52,74]. In agreement with this, disruption of RdDM often led to severe reproductive abnormalities, including but not limited to pollen defects, low seed setting, seed size alterations, etc.

In rice, the mechanisms by which RdDM recognizes TEs and repeats in a tissue-specific manner remains unclear. Our study focuses on the roles of OsCLSY3 and OsCLSY4, two upstream putative chromatin remodelers of the RdDM pathway.

Among the three CLSYs, both OsCLSY3 and OsCLSY4 are expressed in endosperm tissues. In endosperm, imprinted OsCLSY3 expresses at higher level than OsCLSY4, while OsCLSY1 is restricted to the embryo and early growth stages. In panicle tissue, OsCLSY3 is majorly bound to LTR TEs as well as other genomic regions and targeted them for RdDM [32]. This regulation was important for the expression of imprinted genes and seed development-associated genes. In this work, we show ubiquitously expressed OsCLSY4 is also a major player in rice endosperm development, targeting specific TEs and repeats different from those targeted by OsCLSY3 (Fig 8). We found both osclsy3-kd and osclsy4-kd lines had smaller seeds but there were also many phenotypic differences in seeds between the kd lines.

It is important to delineate redundant and non-redundant functions of these CLSY proteins that promote endosperm development and imprinting in rice. As shown in *Arabidopsis*, our data reveal that OsCLSYs majorly act non-redundantly in endosperm but there were a large number of loci in which both function in a redundant manner. Both CLSYs act redundantly in the siren loci and associated genes. Interestingly, 23–24 nt sRNAs were much more reduced in osclsy4-kd than osclsy3-kd which further suggested that ubiquitously expressed OsCLSY4 might be the predominant CLSY in rice, similar to the functions of AtCLSY1 in *A. thaliana*. Recent studies indicated that a large number of maternal sRNAs accumulated in the endosperm originated from surrounding maternal tissues [49,50,52]. Unlike *OsCLSY3*, the *OsCLSY4* is expressed in the surrounding tissues in addition to endosperm and this might aid in the accumulation of sRNAs in endosperm from surrounding maternal tissues. As reported in *Arabidopsis* endosperm [20], rice endosperm also produces a large number of 21–22 nt sRNAs that were under the control of both OsCLSY3 and OsCLSY4. Since the genes associated with siren

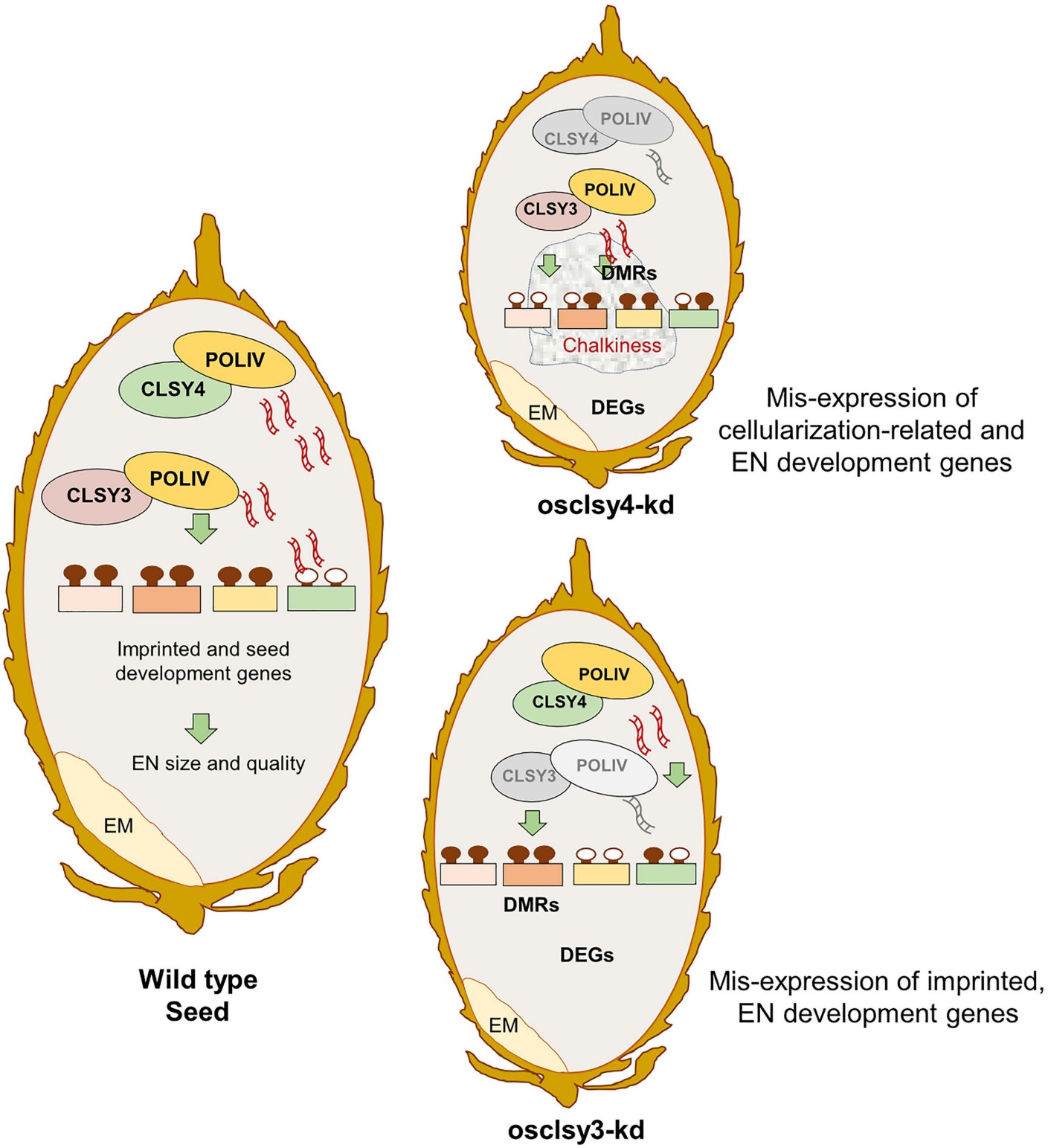

**Fig 8. OsCLSY4 is also a crucial regulator of rice endosperm development.** OsCLSY4 and OsCLSY4 both are important for proper endosperm development in rice. Both the kd lines showed smaller endosperm, but only osclsy4-kd showed grain-chalkiness phenotype. Both rice CLSYs control many unique loci as well as a few shared loci in EN. Those sRNAs direct DNA methylation (filled lollypop) and induced associated epigenetic regulations

that are crucial for proper endosperm development. Both kd lines show hypomethylation at their dependent loci (unfilled lollypops). EM-embryo, EN-endosperm, DEGs-differentially expressed genes, DMRs-differentially methylated regions.

loci are key regulators of imprinting, triploid block and post-zygotic reproductive barrier, CLSYs might act as crucial players in those processes as well [20,52,74]. Also, sequence variation of OsCLSY3 between members of *Oryza* has been documented [32]. Amino acid variations between both OsCLSY3 and OsCLSY4 among different rice accessions as well their expression levels might differentially regulate TEs and repeats, thereby controlling key agronomically important traits. In *Arabidopsis*, CLSY1 D538E mutation is involved in the regulation of lateral root numbers specifically in low K$^+$ conditions [75].

DNA hypermethylation at specific regions was first reported in *Arabidopsis clsy* single mutants along with expected sRNA-dependent hypomethylation [36]. Presence of hyper-DMRs in *clsy* single mutants indicated that CLSYs regulate global DNA methylation and demethylation balance in the genome. It was postulated that DNA demethylation was possibly defective in *clsy* individual mutants to ensure hypermethylation is maintained. However, expression of ROS1 and other demethylases were unaffected in *Arabidopsis clsy* single mutants. A recent study in maize showed loss of ZmROS1a led to reduced CLSY expression, weaker sRNA, deficient RdDM activity and suggested the presence of a feedback loop [76]. In rice, among the six demethylases, *OsROS1* and *OsDML4* were significantly downregulated in osclsy4-kd endosperm. Interestingly, both these demethylases were documented as crucial players in rice endosperm development [77–79].

In several hyper-DMRs, sRNA-independent hypermethylation was observed and the hypermethylated sites were distinct between individual clsy-kd lines. It indicates a possibility that hypermethylation is not necessarily due to redundancy or competition in CLSY-bound regions by individual CLSYs. Pol IV-suppressed sRNAs that depend on Pol II in rice and *Arabidopsis* led to the production of different-sized sRNAs from diverse genomic regions [40]. Such a regulation required CLSY proteins and Pol IV complex assembly, but not other downstream players including DCLs and AGOs [40]. This suggests that different polymerases compete for specific genomic regions to initiate RdDM through various mechanisms. Blocking the canonical RdDM might promote stronger non-canonical RdDM, resulting in hypermethylation at certain loci [80].

It is also possible that the absence of RdDM in specific regions controlled by individual CLSY proteins might have altered nucleosome positions or chromatin marks to initiate more methylation through their properties as chromatin remodelers. For example, FACT complex protein SSRP1 plays a role in DME-mediated DNA demethylation in *Arabidopsis* endosperm [81,82]. Some of the hypermethylation might not be through sRNAs, but through other sRNA-independent DNA methylation or silencing such as mediated in *ddm1* and *mom1* mutants [83–85]. Future research could investigate the mechanism by which hypermethylation arises in *clsy* mutants.

It is interesting to note that individual CLSYs compared here target same family of TEs without competing for any single TE in endosperm. Since such a specific targeting was essential in regulating gene expression necessary for endosperm development and imprinting, it is unclear how such a preference is initiated and maintained. It would be an interesting area for further investigation to understand how TEs and other features are recognized by the CLSYs.

Similar to osclsy3-kd, osclsy4-kd also showed smaller endosperm but in osclsy4-kd, the seeds were slightly wider than PB1. Many known endosperm-development associated genes such as OsBZR1, GW6, OsYUCCA11, NF-YC12, OsDML4, AWN1 and RAG2 were downregulated in osclsy4-kd. Interestingly, osclsy4-kd seeds showed grain chalkiness phenotype which was not observed in osclsy3-kd. This phenotype was also observed in OsCLSY3 OE seeds where OsCLSY3 expression was more than control WT tissues [32]. This suggests that higher expression of OsCLSY3 beyond a threshold result in chalkiness possibly by altering expression of imprinted, seed development genes or metabolic pathway-related genes. Altered seed size, endosperm quality and developmental timing due to altered dose of expression of imprinted genes is well known among plants. Many well-studied seed size-related and imprinted genes such as Os02g0517531,

Os05g0477200, Os05g0447200, Os02g0682200 Os07g0530400, Os05g0490700 were mis-expressed in OsCLSY3 OE, osclsy4-kd and osclsy3-kd endosperm tissues (S9 and S10 Data). From these observations it appears that proper ratio of OsCLSY3 and OsCLSY4 is absolutely important to maintain proper endosperm development, especially the cellularization timing.

Normal rice grains contain tightly packed starch granules, whereas chalky grains exhibit loose, spherical granules that reduce milling quality and nutrient value [64]. Chalkiness is a complex trait influenced by environmental stresses such as drought, heat and nutritional stress conditions [64]. The RdDM pathway helps plants respond to many abiotic stresses such as heat, cold, drought, salt and nutrient starvation [86–88]. TEs are well-known to get de-repressed in abiotic stressed conditions to get inserted near genes. Our findings suggest that OsCLSY4-mediated RdDM may act as the important regulator of stress signaling and seed development. We observed a direct correlation between the abundance of sRNAs and the expression of a few grain chalkiness-associated genes in osclsy4-kd. Future research might reveal how each upstream RdDM pathway player, including but not limited to CLSY proteins, might regulate gene expression under stresses that crops encounter routinely.

## Methods

### Rice transformation and plant growth

For generating transgenic rice plants, *Agrobacterium*-mediated transformation was performed as described previously [89,90]. Briefly, 3 weeks old scutellum-derived calli from PB1 line (*O. sativa indica*) were used. Calli were infected with *A. tumefaciens* strain LBA4404 having *vir* helper plasmid pSB1 (pSB1 carries extra copies of *vir* genes) and the binary plasmids of interest. Hygromycin antibiotic was used as a selection marker. The regenerated transgenic seedlings were maintained in a growth chamber at 23°C with 16h/8h light/dark cycle at 70% relative humidity (RH). The plants were subsequently transferred to a greenhouse.

### Vector design and construction

The amiR that targets OsCLSY4 mRNA was designed using the WMD3 web tool [91,92], with the stringent criteria for robust amiR generation as described previously [45]. The amiR-precursors were synthesized by GeneArt (Thermofisher). The amiR sequence was cloned into pCAMBIA1300 under maize Ubiquitin promoter (P:*Zm*Ubi) with *hygromycin phosphotransferase (hph)* gene as a selection marker. The construct was verified by restriction enzyme-based analysis followed by sequencing. Construct was mobilized into *Agrobacterium* strain LBA4404 (pSB1), and mobilization was verified by PCR analysis.

### Phenotyping of transgenic plants

Phenotypes of transgenic plants such as plant height, panicle length and number of spikelets per panicle were measured using (n > 6) mature plants grown for about 4 months. Phenotypes of roots and shoots of transgenic plants were measured using (n > 8) media-grown 10-day-old seedlings. Images of rice seeds and endosperm from rice lines were obtained using Lecia S8APO stereomicroscope and Nikon camera. For statistical analysis, paired *t*-test was used.

### RNA extraction and RT-qPCR

Total RNA extraction from rice tissues was performed using TRIzol Reagent (Invitrogen) as per manufacturer's instructions. For endosperm tissue, RNA isolation was performed as described earlier [32,93]. RT-qPCR was performed for the expression of OsCLSY4 and other genes. First-strand cDNA was synthesized from 1.0 µg of total RNA using Thermo Scientific RevertAid First Strand cDNA Synthesis Kit, as per manufacturer's instructions.

qPCR was carried out with Solis Biodyne - 5x HOT Firepol Evagreen qPCR Master Mix. As internal control, *OsActin* (Os03g0718100) and *Glyceraldehyde-3-phosphate dehydrogenase* (*OsGAPDH)* (Os04g0486600) were used. RT-qPCRs were performed at least three times using the BioRad CFX system. Primers used for analysis are provided in S3 Table.

### DNA methylation analyses

Total DNA was isolated from 20 day-old endosperm using CTAB method [94]. Equal amount of DNA was sheared to produce 350 bp fragments using ultrasonication (Covaris). Bisulfite conversion was performed using EZ DNA Methylation-Gold Kit as per the manufacturer's instructions. The libraries were constructed using the IDT xGen Methyl-Seq Lib Prep (Catalog no-10009860) and sequenced using NovaSeq 6000 (2X 100 bp mode). Obtained reads were trimmed using Trimmomatic after quality checking [95]. The reads were aligned to the IRGSP1.0 genome using Bismark aligner tool with default parameters [96]. DNA methylation status was extracted and coverage reports were generated using Bismark tools. The CG, CHG and CHH reports were made from the CX report file. The metaplots and heatmaps were generated using the ViewBS methylation package [97].

For the targeted BS sequencing, total DNA was isolated from 60-day-old leaf tissue using the CTAB method. About 200–400 ng of DNA was treated using EZ DNA Methylation-Gold Kit (Zymo Research). The treated DNA was used for PCR as a template. Targeted regions were amplified by JumpStart Taq DNA Polymerase (Sigma). The PCR products were deep sequenced on NovaSeq 6000 (2X 100 bp mode) platform. The obtained reads were quality checked and trimmed using cutadapt [98] and aligned to create genome (target sites) using Bismark aligner tool [96]. The obtained results are analyzed using methylation package ViewBS [97]. Primers used for analysis are provided in S3 Table. The bisulfite conversation efficiencies ranged from 94-96%, slightly below the ideal > 99% threshold, but within the acceptable range for plant WGBS data indicating that methylation estimates remain reliable with minimal bias (S4 Table).

### sRNA library preparation and differential expression analyses

Total RNA extraction from endosperm and sRNA library preparations were carried out as described previously [40]. After checking the quality of obtained reads, the adaptors were trimmed by UEA sRNA Workbench [99]. The filtered sRNAs were classified into 21–22 nt and 23–24 nt reads, and aligned using Bowtie -v 1 -m 100 -y -a --best --strata [100]. The sRNAs loci were identified using Shortstack with the following parameters: --nohp --mmap f --mismatches 1 -mincov 5 rpmm [101]. CLSY-dependent sRNA loci were identified by bedtools intersect [102]. For quantifying the sRNA abundance from transposons, and siren loci, bedtools multicov was used to obtain raw abundance and then normalized to RPKM values [102,103]. To plot the data, inverse hyperbolic sine scale (ASINH) was used. Obtained values were plotted as box plots using custom R scripts in ggplot2 [104]. The venn diagrams were generated by intervene online tools [105]. For publicly available datasets sRNAs loci were identified using Shortstack with following parameters: --nohp --mmap f --mismatches 1 -mincov 2 rpmm [101]. TE enrichment was accessed by calculating the base pair (bp) overlap of CLSY-dependent loci with each TE class relative to randomly shuffled loci. Enrichment was defined as the ratio of overlapped bps to total bps of each TE class in the genome.

### RNA-seq and analyses

RNA-seq was performed using 20-day-old endosperm tissues as source tissue. Poly(A) enrichment was performed before the library preparation using NEBNextUltraII Directional RNA Library Prep kit (E7765L) as per manufacturer's instructions. The obtained libraries were sequenced in NovaSeq 6000 (2X 100 bp mode) platform. The obtained reads were adapter trimmed using Trimmomatic after the quality checking [95] and aligned to IRGSP1.0 genome using HISAT2 with default parameters [106]. Cufflinks was used to perform differential gene expression

(DEG) analyses and statistical testing [107]. Volcano plots were generated for DEGs using custom R scripts with a p-value cut-off of less than or equal to 0.05 and the absolute $\log_2$ (fold change) expression cut-off of more than or equal to 1.5. For quantifying the expression of genes, bedtools multicov [102,103,108] was used to obtain raw abundance and then normalized to RPKM values. DEseq2 was used to identify $\log_2 1.5$ fold change upregulated and downregulated genes with the p-value cut-off of less or equal to than 0.05 [109]. These values were plotted as box plots and heatmaps using custom R scripts in ggplot2 [104]. For publicly available datasets, $\log_2 1.0$ fold change upregulated and downregulated genes with the p-value cut-off of less than or equal to 0.05 were identified by DEseq2 [109].

## GO analysis

The GO analysis was performed using ShinyGO v0.75 platform [110]. The gene IDs were used from RAPDB. The biological processes with FDR cut-off of p-value-0.05.

## SEM imaging

For scanning electron microscope (SEM) imaging, rice endosperms were collected (25 DAP) and fixed in 16% formaldehyde, 25% glutaraldehyde and 0.2 M cacodylate buffer for 12–16 h. The samples were rinsed with double distilled water and dehydrated in a series of ethanol (30%,50%,70% and 100%) passages and dried in critical point drying (CPD, Leica EM CPD300). After CPD, endosperms were cut by a sharp razor blade. Samples were coated with gold, and the images were obtained using a Carl Zeiss scanning EM (Scanner SE2) at an accelerating voltage of 2 kV-4 kV as described before [40,111].

## Germination assay

The germination assay was performed as described previously [112]. A total of 20 seeds of different genotypes (n = 5) were imbibed on the wet filter paper in 5 cm diameter petri-plates. In all the plates, 4 ml of single distilled water was used to wet the filter papers and seeds were germinated at 25°C for 7 days in the dark. Experiments were repeated twice independently. The germination assay was also performed in MS media with surface sterilized seeds of different genotypes.

## Pollen staining

Pollen viability test was performed using $I_2$-KI staining solution containing 0.2% (w/v) $I_2$ and 2% (w/v) KI as described [111,113,114]. Anthers from six spikelets of mature panicles were collected in 200 µl of solution one day before the fertilization. Mechanical shearing was used to release pollens into the solutions. After 10 min, viable pollen grains were counted under the bright-field microscope (OLYMPUS BX43). Before imaging, anther and other debris were removed carefully. Dark stained and round pollen grains were considered as viable, while very light blue and distorted pollens were considered as non-viable [40,115].

## Statistics and reproducibility

No data were excluded from the analysis. All SEM images were performed at least twice with different samples. Statistical analysis was performed using two-tailed paired Student's *t*-test, Fisher's exact test or two-sided Wilcoxon test to determine differences between two groups. Statistical analyses were performed using excel and R studio. Details such as the number of replicates, the level of significance and sample sizes for RT-qPCR, RNA analysis, sequencing, phenotyping were mentioned in the corresponding figure legends, text and S1–S10 Data. A full list of primers, probes, sequences and other details are available in S1–S4 Tables.

## Supporting information

**S1 Fig. Phenotypes of osclsy4-kd plants.**
(PDF)

**S2 Fig. Endosperm-sRNAs are regulated by both OsCLSYs.**
(PDF)

**S3 Fig. OsCLSYs regulate expression of 21–22 nt sRNAs in endosperm.**
(PDF)

**S4 Fig. Genomic distribution and overlap of OsCLSY3- and OsCLSY4-dependent sRNA loci with different genomic features.**
(PDF)

**S5 Fig. DNA methylation in an RdDM loci regulated by OsCLSY4 in leaf.**
(PDF)

**S6 Fig. OsCLSYs regulate DNA methylation in endosperm non-redundantly.**
(PDF)

**S7 Fig. OsCLSY4 regulates expression of protein coding genes.**
(PDF)

**S8 Fig. OsCLSY4 targets different regions for sRNA production in seedling tissue.**
(PDF)

**S1 Table. Details of high-throughput genomics data generated in this study.**
(DOCX)

**S2 Table. Details of high-throughput genomics data obtained from publicly available datasets.**
(DOCX)

**S3 Table. List of oligos and probes used in this study.**
(DOCX)

**S4 Table. Table showing methylation non-conversion rate.**
(DOCX)

**S1 Data. OsCLSY4-dependent sRNA loci.**
(XLSX)

**S2 Data. OsCLSY3-dependent sRNA loci.**
(XLSX)

**S3 Data. Unique and common OsCLSY3 and OsCLSY4 loci.**
(XLSX)

**S4 Data. OsCLSY3 DMRs in Endosperm.**
(XLSX)

**S5 Data. OsCLSY4 DMRs in Endosperm.**
(XLSX)

**S6 Data. Expression of 23–24nt sRNAs from siren loci.**
(XLSX)

**S7 Data. OsCLSY4 and OsCLSY3 DEGs in endosperm.**
(XLSX).

**S8 Data. OsCLSY4-dependent 23–24 nt sRNA loci in seedling.**
(XLSX)

**S9 Data. OsCLSY4 DEGs in seedling.**
(XLSX)

**S10 Data. DEGs OsCLSY3 OE Endosperm.**
(XLSX)

## Acknowledgments

We thank lab members for comments and suggestions. We thank Prof. K. Veluthambi for *Agrobacterium* strains, PB1 seeds and binary plasmids. We thank genomics, electron microscopy, IT, radiation, greenhouse, and lab kitchen facilities at the NCBS.

## Author contributions

**Conceptualization:** Avik Kumar Pal, Padubidri V. Shivaprasad.

**Data curation:** Avik Kumar Pal, Riju Dey, Padubidri V. Shivaprasad.

**Formal analysis:** Avik Kumar Pal, Saumyapriya Rana, Riju Dey, Padubidri V. Shivaprasad.

**Funding acquisition:** Padubidri V. Shivaprasad.

**Investigation:** Avik Kumar Pal, Riju Dey, Padubidri V. Shivaprasad.

**Methodology:** Avik Kumar Pal, Saumyapriya Rana, Riju Dey.

**Project administration:** Padubidri V. Shivaprasad.

**Resources:** Avik Kumar Pal, Padubidri V. Shivaprasad.

**Software:** Avik Kumar Pal, Padubidri V. Shivaprasad.

**Supervision:** Padubidri V. Shivaprasad.

**Validation:** Avik Kumar Pal, Padubidri V. Shivaprasad.

**Visualization:** Avik Kumar Pal, Padubidri V. Shivaprasad.

**Writing – original draft:** Avik Kumar Pal, Padubidri V. Shivaprasad.

**Writing – review & editing:** Avik Kumar Pal, Padubidri V. Shivaprasad.

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
