## [Decision Letter · Decision Letter 0]

25 Jun 2025

PGENETICS-D-25-00277

Loss of function of chromatin remodeler OsCLSY4 leads to RdDM-mediated mis-expression of endosperm-specific genes affecting grain qualities

PLOS Genetics

Dear Dr. Shivaprasad,

Thank you for submitting your manuscript to PLOS Genetics. After careful consideration, we feel that it has merit but does not fully meet PLOS Genetics's publication criteria as it currently stands. Therefore, we invite you to submit a revised version of the manuscript that addresses the points raised during the review process.

Please submit your revised manuscript within 30 days Jul 25 2025 11:59PM. If you will need more time than this to complete your revisions, please reply to this message or contact the journal office at plosgenetics@plos.org. Please include the following items when submitting your revised manuscript:

We look forward to receiving your revised manuscript.

Kind regards,

John M. Greally, D.Med., Ph.D.

Section Editor

PLOS Genetics

John Greally

Section Editor

PLOS Genetics

Aimée Dudley

Editor-in-Chief

PLOS Genetics

Anne Goriely

Editor-in-Chief

PLOS Genetics

**Journal Requirements:**

At this stage, the following Authors/Authors require contributions: Avik Pal, Saumyapriya Rana, Riju Dey, and Padubidri V Shivaprasad. Please ensure that the full contributions of each author are acknowledged in the "Add/Edit/Remove Authors" section of our submission form.

The list of CRediT author contributions may be found here: https://journals.plos.org/plosgenetics/s/authorship#loc-author-contributions

https://journals.plos.org/plosgenetics/s/submission-guidelines#loc-parts-of-a-submission

- ® on pages: 18, and 20

- TM on pages: 19, and 20.

Potential Copyright Issues:

i) Please confirm (a) that you are the photographer of 1C, 1D, 1F, 1G, and S1C, or (b) provide written permission from the photographer to publish the photo(s) under our CC BY 4.0 license.

ii) Figure 8. Please confirm whether you drew the images / clip-art within the figure panels by hand. If you did not draw the images, please provide (a) a link to the source of the images or icons and their license / terms of use; or (b) written permission from the copyright holder to publish the images or icons under our CC BY 4.0 license. Alternatively, you may replace the images with open source alternatives. See these open source resources you may use to replace images / clip-art:

7) Thank you for stating 'All raw and processed sequencing data generated in this study have been submitted to the NCBI Gene Expression Omnibus (GEO; https://www.ncbi.nlm.nih.gov/geo/) under accession numbers GSE290279. Other publicly available data used in this study can be found under GSE229961, GSE180457, GSE138705, GSE130168, GSE229959, GSE215853, , GSE130166, GSE215854, , GSE158710, GSE131319. The reads were aligned to IRGSP1.0 genome. Source data are provided with this paper. To review GEO accession GSE290279: Go to https://www.ncbi.nlm.nih.gov/geo/query/acc.cgi?acc=GSE290279 Enter token cxcvqyigprcjvol into the box' Please note that, though access restrictions are acceptable now, your entire minimal dataset will need to be made freely accessible if your manuscript is accepted for publication. This policy applies to all data except where public deposition would breach compliance with the protocol approved by your research ethics board. If you are unable to adhere to our open data policy, please kindly revise your statement to explain your reasoning and we will seek the editor's input on an exemption.

8) Please send a completed 'Competing Interests' statement, including any COIs declared by your co-authors. If you have no competing interests to declare, please state "The authors have declared that no competing interests exist". Otherwise please declare all competing interests beginning with the statement "I have read the journal's policy and the authors of this manuscript have the following competing interests"

**Reviewers' comments:**

Reviewer's Responses to Questions

**Comments to the Authors:**

Reviewer #1: This paper follows previous work (from the authors and others) outlining the role of OsCLSY proteins in leaves and the role of OsCLSY3 specifically in endosperm. This manuscript focuses on the role of OsCLSY4 in endosperm and how its function overlaps or not with OsCLSY3. The authors observe that there are both unique and shared sites impacted by these related proteins. Unfortunately, they are unable to directly assess redundancy without a double knockdown mutant. The paper includes analysis of sRNA accumulation (sRNAseq), DNA methylation (whole genome bisulphite sequencing), and transcriptomes (RNAseq). The analyses are routine and include some limited integration of datatypes. The primary conclusion is that although OsCLSY3 is specifically expressed in endosperm it is not the sole CLSY protein functioning in this tissue. This finding will interest researchers focused on the mechanism of RdDM, but does not offer deep insights into the process.

Suggestions for improvement:

The manuscript is a bit sloppy and could use careful proof-reading on top of editing for clarity and concision. It rambles quite a bit in the introduction and discussion and it is sometimes difficult to find the important conclusions in the results section. The discussion section is highly speculative in places. For example, “Amino acid variations between both OsCLSY3 and OsCLSY4 between different rice accessions as well their expression variations might differentially regulate TEs and repeats thereby controlling key agronomically important traits” is not based on any particular evidence presented in the paper. The discussion is also quite long and includes a lot of extra detail (eg, explaining the chalky phenotype of endosperm).

Please check reference numbering. For example, at the top of pg 5, I believe this should be Ref 32, not 31.

The first paragraph of the results is redundant with Pal et al 2024, which includes expression data, protein structure, etc. Novel points (eg, sequence identity comparison to RMR1) are trivial. This would be better if integrated into the introduction.

How are the sRNA abundance plots (Fig 2A, S2A, and S2B) normalised? It appears that only 5-10% of the sequenced sRNAs are 20-25 nt in length? In Fig S2, panels A and B are also inconsistent. Panel A shows <0.05 as the abundance of 24-nt sRNAs, but panel B shows 0.25. Similarly, 21- and 22-nt sRNAs are 0.01-0.15 in panel A, but approx 0.07 in panel B.

Pg 6: “Among the pools of 21, 22 and 24 nt sRNAs, sRNAs that had 5’ A were specifically reduced in osclsy4-kd” This statement is not supported by the data in Sup Fig 2B, which shows strong reduction in 24-nt siRNAs regardless of first nucleotide.

Pg 7: “many of the OsCLSY4-dependent 23-24 nt sRNAs from TEs overlapped with OsCLSY3-dependent loci 23-24 nt loci” is also unsupported by the data shown. Fig 2G shows two genome regions and does not quantify overlap between OsCLSY3- and OsCLSY4-dependent loci.

Pg 7: “we analysed published sRNA datasets derived from osclsy3-kd endosperm tissues” Please provide a citation for these data and describe which tissue they represent, so that the reader might immediately understand the comparison you are making.

Fig2-3: What are “ASINH” converted RPKM values?

Fig 3H is an uncontrolled experiment. How can the authors determine that there is no “preference for a specific feature of TE” when they don’t know what the genome frequency of these elements are? Same concern for Fig S5B.

Sup Fig S4 is superfluous to this story. DNA methylation in leaves of clsy4 mutants is already published and this paper is focused on endosperm.

There appears to be only a small impact on siren sRNA abundance and almost no effect on DNA methylation in clsy3- or clsy4-kd lines. I would therefore not conclude that “both OsCLSYs play a major role in their regulation.” In Arabidopsis siren sRNAs are almost completely eliminated in clsy3 mutants; there is clearly something different happening in rice, according to this data.

Pg 15: “the genes associated with siren loci are key regulators of imprinting, triploid block and post-zygotic reproductive barrier” What is the evidence for this statement?

Data supporting the quality of libraries should be included as a supplement. Number of reads, number mapping to the genome, and (critically!) bisulphite conversions values for each library.

Reviewer #2: In this manuscript, Pal and colleagues studied the role of CLASSY (CLSY) proteins in rice small RNA function, TE silencing and gene regulation. In particular this work seeks to tease apart distinct roles of two related proteins, OsCLSY3 from OsCLSY4. The authors published extensively on the molecular and regulatory functions of OsCLSY3 in Pal et al. 2024 Nat Commun. The current manuscript is focused on a narrower set of analyses using artificial microRNA knockdown in the indica rice line Pusa Basmati-1 (PB1), producing osclsy4-kd lines deficient for OsCLSY4. Endosperm sRNAs are globally reduced in these osclsy4-kd lines compared to PB1. The study includes the standard epigenomics approaches: RNA-seq, sRNA-seq and whole-genome bisulfite sequencing, implicating both OsCLSY3/4 proteins in regulating certain TEs and genes.

As could be expected based on studies in Arabidopsis, loss of OsCLSY4 function affects a subset of 23/24-nt siRNA loci and associated RNA-directed DNA methylation (Zhou et al 2018 Nat Genet.). This follows directly from known functions of CLSY proteins in Pol IV recruitment in flowering plants, including Arabidopsis and rice (Zhou et al 2018; Pal et al. 2024). The specialized function of CLSYs as parental dosage regulators in the rice endosperm has key biological and agricultural implications, which was nicely demonstrated by the authors.

I am concerned that recent publications, and the current manuscript, are creating inconsistent names for CLSY proteins in rice. Alternative naming arose in the context of a genetic screen, which referred to FEM2, FEL1 and FEL2 proteins (Xu et al, 2024 Plant Physiol.) rather than simply calling them rice homologs to Arabidopsis CLSYs (Smith et al. 2007). Moreover, the rice CLSYs were already identified and named in detailed phylogenetic work by Trujillo et al. 2018 Mol Biol Evol. Inconsistent names were again used in Pal et al. 2024 (i.e., OsCLSY3(Pal), OsCLSY4 (Pal)). What is called “OsCLSY3” here and in Pal et al. 2024 corresponds to OsCLSY5(Trujillo). By contrast, what is called “OsCLSY4” here and in Pal et al. 2024 was called OsCLSY3(Trujillo) in 2018. The authors should properly cite and discuss this previous phylogenetic work, which first differentiated the Poaceae-specific CLSY5(Trujillo) subfamily from its more broadly conserved CLSY3(Trujillo) paralogs.

Furthermore, the quite specific developmental regulation of OsCLSY3(Pal) in rice--expressed in young endosperm 15 days after pollination--is more consistent with the subfunctionalized role symbolized in the name OsCLSY5(Trujillo). On this account, the broad expression and redundant function of OsCLSY4(Pal), described in the current manuscript, seems more consistent with the original OsCLSY3(Trujillo) name, especially because this protein groups phylogenetically with CLSY3 proteins from species outside of the Poaceae, which all lack the CLSY5 subfamily. Therefore, when the authors state (page 11), “It is important to note that the homologs of rice CLSY3 and CLSY4, i.e., AtCLSY3 and AtCLSY4, also regulate siren loci expression in ovules and endosperm indicating possible conserved functions of this clade of CLSY members,” the distinct OsCLSY3(Pal)=OsCLSY5(Trujillo) subfamily is obscured and misassigned.

Later in the Discussion, the authors state that, “23-24 nt sRNAs were much more reduced in osclsy4-kd than osclsy3-kd which further suggested that ubiquitously expressed OsCLSY4 might be the predominantly involved CLSY in rice, similar to AtCLSY1.” In my view, this simply follows from the result that OsCLSY4(Pal)=CLSY3(Trujillo) has broader expression than the more specialized OsCLSY3(Pal)=OsCLSY5(Trujillo) protein, which evolved more recently and shows more limited expression in the endosperm. Certainly, in 18-d-old seedlings, mutations the disrupt OsCLSY4(Pal)=CLSY3(Trujillo) certainly led to the most dramatic loss of 24-nt siRNAs (Xu et al, 2024 Plant Physiol.). However, I do not think that this makes OsCLSY4(Pal) similar to AtCLSY1 in any deep functional sense, given that AtCLSY1 is not involved in siren siRNA biogenesis in Brassicaceae species.

In summary, I recommend that the authors try to make a better accounting of the evolutionary origins of OsCLSY4(Pal)=CLSY3(Trujillo) and the unfortunate recent shift in nomenclature. Much of the data that the authors present here, and presented previously in Pal et al. 2024, make more sense in light of this particular protein being closely related to Arabidopsis CLSY3 and Maize CLSY3/RMR1 (Hale et al. 2007; Trujillo et al. 2018), rather than the alternative paralogs.

Minor issues and corrections:

Introduction, paragraph 1: “…the rapid evolution of angiosperms often considered as an abominable mystery”. Consider rephrasing here. In English, “abominable” is reserved for terrifying, unpleasant, or disgusting things. A common example is the “abominable snowman”.

Throughout the manuscript: the definite article “the” is frequently missing. Here are some examples on page 2:

(1) “Unlike gymnosperms, majority of the angiosperm seeds contain a triploid endosperm…”  “Unlike gymnosperms, _the_ majority of the angiosperm seeds contain a triploid endosperm…”

(2) “Incorporation of paternal genome in the nutrient tissue endosperm…”  “Incorporation of _the_ paternal genome in the nutrient tissue endosperm…”

(3) “In cereal seeds, starch-filled endosperm is the major food source for humans and

other animals”  “In cereal seeds, _the_ starch-filled endosperm is the major food source for humans and other animals”

(4) “Multiple studies in model plant Arabidopsis indicated that”  “Multiple studies in _the_ model plant Arabidopsis indicated that”

Introduction, page 2: The authors overuse the verb “regulate” to describe the activities of DNA methyltransferases. METHYLTRANSFERASE1 (MET1) catalyzes 5-methylcytosine maintenance at CG dinucleotides, rather than just “regulating” CG methylation. The same problem arises when the authors state that CMT3, CMT2 or DRM2 “regulate” DNA methylation in particular contexts, because they do not specify what they mean by the verb. “Regulate” suggests the enzymes act in a dynamic process involving differential targeting, developmental timing, and/or responsiveness to particular environmental cues. This is partially true, but it is not really the point of the paragraph. Finally, DRM2 can mediate de novo DNA methylation in all sequence contexts, not just CHH methylation.

Results, page 5: “The RdDM pathway plays a crucial role in pollen viability and reproductive success in plants and several mutants in this pathway, such as nrpd1a, clsy3, several AGO genes exhibit pollen sterility [21,32,40].” In Arabidopsis thaliana and Capsella rubella species, this “nrpd1a” mutation should be referred to as “nrpd1”, because the gene mutation once called “nrpd1b” is now conventionally called “nrpe1”. NRPD1 and NRPE1 are the largest subunits of quite distinct RNA polymerases, Pol IV and Pol V, not redundant a / b paralogs forming the same enzyme. Also, “several AGO genes” is at odds with the logical flow in this sentence. The authors were referring to mutants, not genes, at the outset of their list.

**Have all data underlying the figures and results presented in the manuscript been provided?**

Reviewer #1: Yes

Reviewer #2: Yes

PLOS authors have the option to publish the peer review history of their article (what does this mean? ). If published, this will include your full peer review and any attached files.

**Do you want your identity to be public for this peer review?** For information about this choice, including consent withdrawal, please see our Privacy Policy .

Reviewer #1: No

Reviewer #2: No

**Figure resubmission:**
---

## [Decision Letter · Decision Letter 1]

19 Sep 2025

PGENETICS-D-25-00277R1

Loss of function of chromatin remodeler OsCLSY4 leads to RdDM-mediated mis-expression of endosperm-specific genes affecting grain qualities

PLOS Genetics

Dear Dr. Shivaprasad,

Thank you for submitting your manuscript to PLOS Genetics. After careful consideration, we feel that it has merit but does not fully meet PLOS Genetics's publication criteria as it currently stands. Therefore, we invite you to submit a revised version of the manuscript that addresses the points raised during the review process.

My concern is that Reviewer 1 continues to have a number of concerns that do not appear to have been adequately addressed in the revised manuscript. These concerns look to be reasonable and will need to be fully addressed in any revised submission.

Please submit your revised manuscript within 60 days Nov 18 2025 11:59PM. If you will need more time than this to complete your revisions, please reply to this message or contact the journal office at plosgenetics@plos.org. Please include the following items when submitting your revised manuscript:

We look forward to receiving your revised manuscript.

Kind regards,

John M. Greally, D.Med., Ph.D.

Section Editor

PLOS Genetics

John Greally

Section Editor

PLOS Genetics

Aimée Dudley

Editor-in-Chief

PLOS Genetics

Anne Goriely

Editor-in-Chief

PLOS Genetics

**Journal Requirements:**

Please ensure that the CRediT author contributions listed for every co-author are completed accurately and in full.

At this stage, the following Authors/Authors require contributions: Avik Pal, Saumyapriya Rana, Riju Dey, and Padubidri V Shivaprasad. Please ensure that the full contributions of each author are acknowledged in the "Add/Edit/Remove Authors" section of our submission form.

The list of CRediT author contributions may be found here: https://journals.plos.org/plosgenetics/s/authorship#loc-author-contributions

**Reviewers' comments:**

Reviewer's Responses to Questions

**Comments to the Authors:**

Reviewer #1: I am disappointed that the authors’ have made few changes in response to the reviewers’ suggestions. In particular, there continue to be many problems in the manuscript, including issues that were raised in the first round of review.

- The authors cannot state CLSY3 and CLSY4 “do not show any preference for a particular type of TE” without comparison to genome averages in Fig 3H. They can conclude that CLSY3 and CLSY4 have *the same* preferences for genomic features, but they have not defined these preferences. For example, if 80% of the OsCLSY3 targets were LINE elements and 80% of the OsCLSY4 targets were also LINE elements, it would be obvious that both proteins have a preference for LINEs, because the genome is not 80% LINE element. But I cannot judge whether 5-10% MITEs and ~30-40% LTRs is enrichment or depletion of these elements relative to the genome. I can only conclude that there is no difference between CLSY3 and CLSY4 preference.

- I do not know how the authors conclude that “among the pools of 21, 22, and 24 nt sRNAs, sRNAs that had a 5’ A were specifically reduced in osclsy4-kd” (Sup Fig 2B). It is clear from the revised figure that sRNAs with a 5’A (brown bars) are slightly reduced at the 21-nt size, reduced by about half in the 22-nt size, and reduced by at least 75% in the 24-nt class. Clearly sRNAs with a 5’ A were not *specifically* reduced in osclsy4-kd. This is such a minor part of the published story I’m not even sure why it is included, but by misrepresenting their data (even after correction in review), it makes everything look suspect.

- Similarly, there is no explanation for the inconsistency between Sup Fig 2A and 2B. Both of these plots show “sRNA abundance (Million)” while the response to reviewers says that it is “normalized sRNA abundance (Millions)”. Regardless of the title, I assume the normalization was the same in both plots, yet the scales are quite different. There isn’t even a description of what these are normalized against in the manuscript. As someone in the field, I would be uncomfortable citing this research, since a reader can’t even infer how such analysis was done.

- Reviewer 2 makes an excellent point about CLSY3(Pal) being a Poaceae-specific duplication (CLSY5 in Trujillo) and CLSY4 resembling the ancestral CLSY3/4. While the revised text is a little more clear about the different naming systems, the idea of a Poaceae-specific duplication is not included. Given the unique endosperm of the grass family and the endosperm-specific expression of CLSY3 (CLSY5), this would seem to be an important point.

- Thank you for the information about “ASINH” normalization. I assume ASINH is “Hyperbolic arcsine scaling”. I recommend that you define this somewhere in the manuscript, since it is not clear to this reader and probably many PLoS Genetics readers.

- The bisulfite conversion data (and other HTS data information) is essential and should be provided in the manuscript, not just to the reviewers. I also am puzzled by the addition of a conversion rate and a non-conversion rate. I have only seen this applied in mammalian systems where the conversion rate analyzes only CG dinucleotides (which are not expected to convert in mammalian cells), while non-conversion rate applies to non-CG sites. If that were applied to the chloroplast and mitochondrial sequences, then these samples have 4-5% non-conversion (bisulfite failure) and are therefore acceptable, but borderline. This is important information for readers and should be included, but the authors must only report the meaningful rate - the rate of non-converted cytosines.

Reviewer #2: In this revision, Pal and colleagues significantly improved their manuscript, which focuses on functional differences between the CLASSY (CLSY) proteins they call CLSY3 and CLSY4 in rice. The authors addressed most of the reviewers’ main concerns. The nomenclature problem, in which inconsistent names were assigned to proteins variously called “CLSY3”, “CLSY4” or “CLSY5” in past studies, remains quite difficult to resolve in the context of the current study. The authors have explained the rationale for their alternative choices. I do recommend that they re-read the manuscript, however, to correct some remaining English language issues.

Minor corrections:

Abstract: “RNA-directed DNA methylation (RdDM) sequence-specifically targets transposable elements (TEs)…” I recommend that they rephrase this sentence to, “In plants, RNA-directed DNA methylation (RdDM) targets transposable elements (TEs) and repeats in a sequence-specific manner, often including tissue-specific factors” …or similar for better readability.

Pg3, paragraph 2: “Although, clsy1,2 and clsy3,4 double mutants regulated comparatively more number of sRNA loci than single mutants, suggesting some degree of redundancy within the CLSY family proteins [26,27,35]”, should start, “However, clsy1,2 and clsy3,4 double mutants affected a greater number of sRNA loci…”

Pg4, paragraph 1: “a clear mechanistic understanding has been eluded,” should be “a clear mechanistic understanding has been elusive.”

Pg5, paragraph 1: “N-terminal intrinsically-disorder regions, …” needs correcting to “N-terminal intrinsically disordered regions, …”

Pg6, paragraph 2: “class I (TEs that replicate in genome by copy paste mechanism) … class II (… cut paste mechanism)”, is typically written as follows, “class I (TEs that replicate in the genome by a copy-and-paste mechanism) … class II (… cut-and-paste mechanism).”

Pg18, paragraph 1: “We did not observe any preference of either CLSYs for specific features”. Grammatical correction: “We did not observe any preference of either CLSY for specific features.”

Generally, it is helpful to include line numbers in reviewer copies to facilitate corrections.

**Have all data underlying the figures and results presented in the manuscript been provided?**

Reviewer #1: Yes

Reviewer #2: Yes

PLOS authors have the option to publish the peer review history of their article (what does this mean? ). If published, this will include your full peer review and any attached files.

**Do you want your identity to be public for this peer review?** For information about this choice, including consent withdrawal, please see our Privacy Policy .

Reviewer #1: No

Reviewer #2: No

**Figure resubmission:**
---

## [Editor Report · Decision Letter 2]

12 Nov 2025

Dear Dr Shivaprasad,

We are pleased to inform you that your manuscript entitled "Loss of function of chromatin remodeler OsCLSY4 leads to RdDM-mediated mis-expression of endosperm-specific genes affecting grain qualities" has been editorially accepted for publication in PLOS Genetics. Congratulations!

Yours sincerely,

John M. Greally, D.Med., Ph.D.

Section Editor

PLOS Genetics

John Greally

Section Editor

PLOS Genetics

Aimée Dudley

Editor-in-Chief

PLOS Genetics

Anne Goriely

Editor-in-Chief

PLOS Genetics

BlueSky: @plos.bsky.social

Comments from the reviewers (if applicable):

**Data Deposition**

http://datadryad.org/submit?journalID=pgenetics&manu=PGENETICS-D-25-00277R2

**Press Queries**

---

## [Editor Report · Acceptance letter]

PGENETICS-D-25-00277R2

Loss of function of chromatin remodeler OsCLSY4 leads to RdDM-mediated mis-expression of endosperm-specific genes affecting grain qualities

Dear Dr Shivaprasad,

We are pleased to inform you that your manuscript entitled " 

Loss of function of chromatin remodeler OsCLSY4 leads to RdDM-mediated mis-expression of endosperm-specific genes affecting grain qualities " has been formally accepted for publication in PLOS Genetics! Your manuscript is now with our production department and you will be notified of the publication date in due course.

With kind regards,

Anita Estes

PLOS Genetics

On behalf of:
